# Large scale purification in semiconductors using Rydberg excitons

Martin Bergen[1], Valentin Walther [2,3,4,5], Binodbihari Panda [1], Mariam Harati[1], Simon Siegeroth [1], Julian Heckötter [1] & Marc Aßmann [1] ✉

Improving the quantum coherence of solid-state systems is a decisive factor in realizing solid-state quantum technologies. The key to optimize quantum coherence lies in reducing the detrimental influence of noise sources such as spin noise and charge noise. Here we demonstrate that we can utilize highly-excited Rydberg excitons to neutralize charged impurities in the semiconductor Cuprous Oxide - an effect we call purification. Purification reduces detrimental electrical stray fields drastically. We observe that the absorption of the purified crystal increases by up to 25% and that the purification effect is long-lived and may persist for hundreds of microseconds or even longer. We investigate the interaction between Rydberg excitons and impurities and find that it is long-ranged and based on charge-induced dipole interactions. Using a time-resolved pump-probe technique, we can discriminate purification from Rydberg blockade, which has been a long-standing goal in excitonic Rydberg systems.

Charge noise results in fluctuating electric fields and random Stark shifts which serve as a decoherence mechanism for exciton and spin states in semiconductors[1–3]. Charged impurities constitute one of the most prominent sources of charge noise in solid-state systems[4–7]. For the special case of zero-dimensional semiconductor quantum systems, such as quantum dots, where only temporal fluctuations matter, a common strategy to overcome this problem consists of illuminating the sample with above-gap excitation[8]. This creates free carriers that saturate the defect which fixes the Stark shift at the position of the localized system[9].

However, for spatially extended systems, such as semiconductor excitons, this approach is insufficient as spatially inhomogeneous Stark shifts persist. Through their optical excitation, excitons are created across the sample and also propagate within their lifetime, thus exploring the electrostatic defect landscape. Internally excited excitons, so-called Rydberg excitons, are extremely polarizable and therefore particularly sensitive to external electric fields. Here, we consider Rydberg excitons in cuprous oxide ($Cu_2O$)[10,11], where excitons with principal quantum numbers up to $n \approx 30$ can be excited[12]. Their linewidths scale as $n^{-3}$, while their polarizability in external fields even

scales as $n^7$ [13], resulting in a strong sensitivity to external fields[14–17]. The predominant defects in natural $Cu_2O$ are deep and positively charged oxygen vacancies with binding energies of several hundred meV[18,19], which can be understood as Coulomb-like static electric potentials in the crystal. The influence of such charged impurities on the Rydberg exciton spectrum in a bulk medium has been studied theoretically in detail for $Cu_2O$[20].

Through their extreme interaction with their electric environment, Rydberg excitons lend themselves as an exquisite spectroscopic probe of charged impurities. The basic interaction mechanism is that the energy of a Rydberg exciton state undergoes a strong Stark shift in the vicinity of a charged impurity. Thus, the absorption attenuation of a spectrally narrow laser beam resonant only with the bare unshifted Rydberg exciton state will signal the presence of charged impurities.

However, this exciton-impurity interaction competes with another spectroscopically relevant effect: Rydberg blockade[10,21,22]. Rydberg blockade describes the excitation blockade of two Rydberg excitons within a blockade radius on the order on one to several micrometers. While a highly successful resource of entanglement and strong correlations[23] with both fundamental[24,25] and technological

[1]Experimentelle Physik 2, Technische Universität Dortmund, D-44221 Dortmund, Germany. [2]ITAMP, Harvard-Smithsonian Center for Astrophysics, Cambridge, MA, USA. [3]Department of Physics, Harvard University, Cambridge, MA 02138, USA. [4]Department of Chemistry, Purdue University, West Lafayette, IN 47907, USA. [5]Department of Physics and Astronomy, Purdue University, West Lafayette, IN 47907, USA. ✉e-mail: marc.assmann@tu-dortmund.de

applications[26], Rydberg blockade is known to lead to absorption attenuation much like the exciton-impurity interaction, and therefore complicates the investigation of impurity effects studied here.

In addition, optical excitation of cuprous oxide can create free charges in the form of a thin electron-hole plasma, which can have a strong impact on the Rydberg exciton system. It was shown that such a plasma leads to a reduction of exciton absorption either via dynamical screening[27] or plasmon scattering[28–30] at densities of about $0.01/\mu m^3$ and higher. However, at low optical excitation powers, in particular at excitation energies below the band gap, i.e. when excitons are excited, an estimation of its density is challenging. Moreover, its impact on the mutual interaction between Rydberg excitons and between Rydberg excitons and impurities is a complex problem and part of an ongoing debate.

In this work, we show how to clearly discriminate the interaction mechanisms of Rydberg excitons, by using a pump-probe technique to resolve the time-dependent dynamics. We can thus uncover the dynamical interaction of Rydberg excitons with charged impurities that goes beyond the static shift model outlined above: Rydberg excitons can break up in a dynamical collision with a charged impurity, whereby the impurity is neutralized. The neutralized impurity can, in turn, no longer inhibit the resonant excitation of Rydberg excitons and hence no longer attenuate absorption. This mechanism already occurs at very low powers and results in an effectively increased absorption of the material. We term this effect "purification", as it offers an ultra-low intensity tool to cancel the effects of charged impurities in a homogeneous fashion.

The paper is structured as follows: We first review the material system Cu₂O and the cw-spectroscopy of Rydberg excitons. We start with a description of the pump-probe scheme and the observed interaction effects. We show that the temporal dynamics of purification are effective on the microsecond time scale. This clearly discriminates it from Rydberg blockade, which we show acts on time scales below the temporal resolution of the experiment. We develop a theoretical model describing the exciton-capture process as well as the time-dependent impurity dynamics. We demonstrate that the measured dynamics follow characteristic scaling laws in dependence on the principal quantum number $n$ and show that they are in agreement with the suggested capture model. Finally, we present an example case of how the purification mechanism may be used to mitigate the detrimental effects of charged impurities in materials without spectrally resolvable Rydberg states.

## Results

### Experiment
Cu₂O is a direct-gap semiconductor, where direct dipole-transitions are suppressed due to the same parity of the upmost valence band and lowest conduction band. However, the excitation of $P$ excitons becomes weakly dipole allowed due to their odd-parity envelope that describes the relative motion of electron and hole. We show an example of a linear transmission spectrum of a thin Cu₂O slab in Fig. 1a, where Rydberg exciton states from $n = 9$ to 22 are visible. The transmission of the laser in resonance with an individual exciton line grows with increasing principal quantum number $n$ until the series of exciton lines transforms into an absorption continuum above the band gap $E_g$.

A typical example for Rydberg blockade in continuous wave (cw) experiments[21] without temporal resolution is presented in Fig. 1b. We plot the differential transmission of a probe beam scanned across the $n = 9$ resonance for cw pumping at $n = 16$, i.e. the difference of the transmitted intensity with and without pump laser $\Delta I = I_0 - I(\text{pump})$. At high pump powers, Rydberg blockade sets in as can be clearly identified by the enhanced transmission of the probe beam, which is a result of a large number of Rydberg excitons with $n = 16$ being present in the crystal. Due to the strong van-der-Waals interaction, each of them shifts the resonance energy of the $n = 9$ exciton state out of the state

linewidth within their blockade volumes. The enhancement of probe beam transmission is directly proportional to the total crystal volume blocked for resonant absorption by this mechanism.

At low pump powers, the probe beam transmission instead becomes reduced and the absorption increases. The origin of this effect has not been identified so far. It has been speculated that this effect is related to stray fields arising from charged impurities. It was assumed that they cause Stark shifts that locally shift the energies of Rydberg exciton states when they carry a net charge, while they do not cause any significant shifts when they are in a neutral state[20]. As a consequence, when impurities change from the charged to the neutral state, the absorption of the resonantly probed exciton line will increase, just as observed in the experiment for low pump powers.

To investigate the origin of purification and its dynamics, we perform a time-resolved relative transmission measurement which consists of a pulsed pump beam and a cw probe beam. Both show narrow spectra of about 1 neV and are set to resonance with different Rydberg exciton states. The probe beam monitors the change in the transmission of the probed state when a pulsed pump beam resonant with a different Rydberg exciton state is present. The relative transmission is the ratio of the time-resolved transmitted intensity of the probe beam when the pump pulse is present to the transmitted probe beam intensity when the pump beam is blocked. For details about the experimental setup, see Methods.

A series of time-resolved relative transmission traces for different pump powers is shown in Fig. 1c. At high pump powers, the relative probe beam transmission becomes enhanced due to Rydberg blockade. Blockade builds up almost instantly when the pump beam arrives and also disappears almost instantly when the pump beam is switched off. This implies that blockade is solely mediated by the presence of Rydberg excitons. Their lifetimes are on the scale of nanoseconds or below[11], so their decay is indeed almost instantaneous on the time scales considered here.

At low pump powers, the relative transmission of the probe beam instead decreases by up to 25%, which we term "purification". This decrease corresponds to an enhancement of absorption by up to 25%. It already occurs at very low pump intensities below $1 \frac{mW}{cm^2}$. The purification dynamics differ significantly from the blockade dynamics. Purification builds up during a time scale $\tau_1$ when the pump beam arrives and decays exponentially during a time scale $\tau_2$ when the pump beam is switched off. These time scales are in the microsecond range and are significantly longer than the lifetimes of Rydberg excitons, so they indicate the presence of another type of long-lived carriers in the system: charged impurities[19,31–33]. Most importantly, the data at intermediate pump densities shows that blockade and purification are independent effects and may happen simultaneously. The characteristic step-like shape of the relative transmission curve observed for a pump intensity of $124 \frac{mW}{cm^2}$ demonstrates exactly that, see Fig. 1d. While the pump beam is present the relative transmission drops to a lower value. When the pump beam is switched off, the signal decays on a long timescale from an even lower amplitude. The measured step-like curve can be separated into two time traces with opposite sign. The blue curve describes blockade that vanishes immediately after the pump pulse. The red and green curves describe the onset and decay of purification, which decays much more slowly. Time-resolved measurements make it possible to consider all effects separately, which is already a significant improvement compared to cw studies. At low powers only purification is observed, but blockade is not taking place. In the following, we focus on this low power region as it allows us to study solely purification. Moreover, we focus on the arrangement $n_{\text{pump}} > n_{\text{probe}}$, since we only observe purification when the pump laser is set to an exciton resonance with a principal quantum number larger than the probe's. This is verified in Fig. 1e, where we show a pump-power series for the combination $n_{\text{pump}} = 10$ and $n_{\text{probe}} = 12$, i.e. $n_{\text{pump}} < n_{\text{probe}}$. Here, the change in relative transmission is always

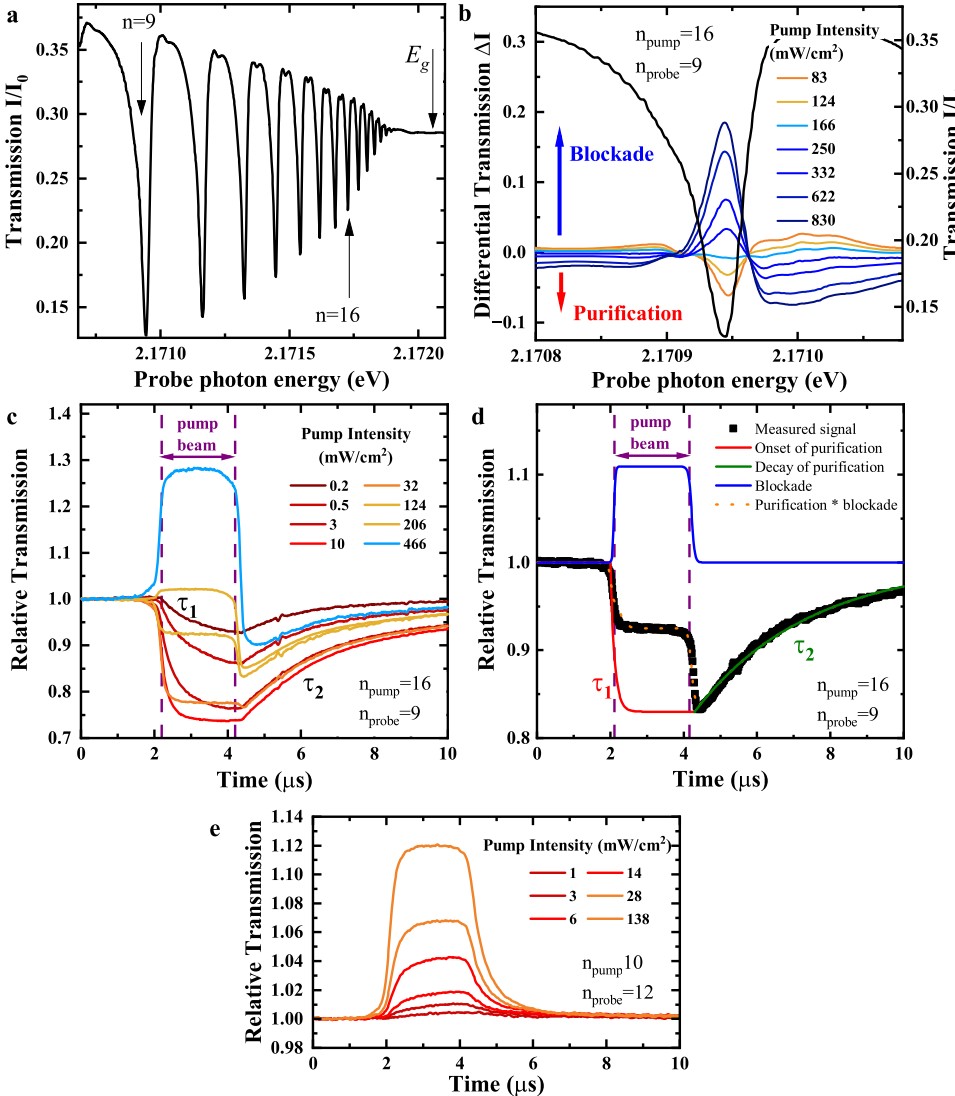

**Fig. 1 | Time-resolved pump-probe spectroscopy. a** Normalized probe laser transmission spectrum of Rydberg excitons from $n = 9$ up to the band gap at 1.3 K. **b** Left axis: Differential transmission $\Delta I = I_0 - I(\text{pump})$ in a cw pump-probe experiment with $n_{\text{pump}} = 16$ and $n_{\text{probe}} = 9$. For low pump intensities the transmission is reduced (purification). For large pump intensities the transmission is enhanced (blockade). Right axis: Norm. transmission around the $n_{\text{probe}} = 9$ resonance for comparison. **c** Pump-intensity series of time resolved relative transmission in a pulsed pump-probe experiment for constant probe power. Purification and blockade may occur simultaneously, but are clearly distinguishable. $\tau_1$ decreases with pump power. The purple dashed lines indicate the duration of the pump pulse. **d** Breakdown of a typical relative transmission time trace for a pump intensity of $124 \frac{mW}{cm^2}$. Upon the arrival of the pump beam indicated by the dashed purple lines, blockade sets in immediately and vanishes immediately at the end of the pump beam. Purification slowly builds up during a time $\tau_1$ and decays on an even slower time $\tau_2$. Considering blockade and purification simultaneously reproduces the step-like structure of the measured relative transmission. **e** Same as b but for $n_{\text{pump}} < n_{\text{probe}}$, where only positive amplitudes are observed.

positive, irrespective of the pump power. A detailed discussion of the role of principal quantum numbers is given in the next Section Model.

## Model

Due to the slow dynamics of the purification process we assume that it corresponds to a change of the average charge state of the impurities mediated by Rydberg excitons. Before analyzing the dynamics in detail, we propose a model to describe the impurity system and the interaction between excitons and impurities:

The density of impurities in our system is $\rho_0$. The probe beam transmission depends on the average density $\rho_{ion}$ of impurities in the charged state. The density of impurities in our system is far below $1 \, \mu m^{-3}$ [20], so the charged impurities can be considered as independent of each other and $\rho_{ion} = \rho_0 p_{ion}$, where $p_{ion}$ is the probability for any single impurity to be in the charged state. In this limit, the problem reduces to finding the dynamics of $p_{ion}$.

We consider several competing processes. A Rydberg exciton hitting a charged impurity may become trapped at a charged impurity and screen it or dissociate and neutralize the impurity, while the remaining charge moves to the sample surface. These processes shift impurities from the charged to a neutral state. On the other hand, also processes turning impurities from a neutral state back towards the ionized state occur. The electric fields arising from the pump and probe light fields may tilt the potential barrier in the impurity, allowing carriers to tunnel away from the impurity. In Rydberg physics, the physical properties that govern these rates show a characteristic scaling with the principal quantum number of the state involved. Accordingly, the scaling laws of these rates point us towards the interaction mechanism between excitons and charged impurities.

We consider $r_{\text{cap}}$ as the rate at which the processes shifting the impurities from the charged to the neutral state take place and $r_{\text{ion}}$ as the rate at which the reionizing processes take place. This is a standard

dynamic equilibrium problem, which has been studied in detail, e.g., for reversible reactions in chemistry[34]. The dynamics of $p_{ion}$ are given by:

$$\dot{p}_{ion} = r_{ion} - (r_{ion} + r_{cap})p_{ion},\qquad(1)$$

which results in the following steady state probability to find an impurity in the charged state:

$$p_{ion,ss} = \frac{r_{ion}}{r_{ion} + r_{cap}}.\qquad(2)$$

When the rates $r_{ion}$ or $r_{cap}$ are suddenly changed at $t = 0$, $p_{ion}$ will develop towards a new steady state. The difference between the old and new steady state is given by $\Delta p_{ion} = p_{ion,ss,new} - p_{ion,ss,old}$ and the dynamics are described by

$$p_{ion}(t) = p_{ion,ss,new} - \Delta p_{ion} e^{-(r_{ion,new} + r_{cap,new})t}.\qquad(3)$$

Interestingly, the time scale of the dynamics always depends on the sum of both rates, irrespective of the sign of $\Delta p_{ion}$. Therefore, any such sudden change will allow us to measure these rates via the time scale $\tau = (r_{ion,new} + r_{cap,new})^{-1}$ of the dynamics.

During our pump-probe experiment, we introduce two sudden changes of the steady state probability. Initially, only the probe beam is present. Although the probe beam intensity is low, it also yields contributions $r_{cap,probe}$ and $r_{ion,probe}$ to the capture and ionization rates and thus defines the initial state of the impurities. When the pump beam arrives, the additional Rydberg excitons yield additional capture and ionization rates $r_{cap,pump}$ and $r_{ion,pump}$ that shift the probabilities towards a more neutral steady state as indicated by the reduced relative transmission. This shift takes place at a rate $\tau_1^{-1} = r_{cap,pump} + r_{ion,pump} + r_{cap,probe} + r_{ion,probe}$. When the pump beam is switched off, only the probe beam is present and the impurities revert back to the initial steady state at a rate $\tau_2^{-1} = r_{cap,probe} + r_{ion,probe}$, which depends only on the probe beam.

For the neutralization process, we may define a capture rate per exciton density, $\Gamma_{cap}(n)$, that depends strongly on the state of the exciton and therefore on its principal quantum number $n$. A larger optical intensity increases the density of excitons created and therefore also $r_{cap}$. We can describe the conversion between the optical intensity $I$ and the created exciton density via a factor $g_0(n) = \alpha(n) T(n)/E$, with the peak absorption $\alpha$ and the lifetime $T$ of the exciton state with principal quantum number $n$. $E$ is its energy. The full capture rate is then given by:

$$r_{cap}(n) = \Gamma_{cap}(n)g_0(n)I.\qquad(4)$$

In a crystal without detrimental effects, the exciton lifetime is limited by radiative or phonon-assisted recombination, both resulting in a $n^3$-scaling of $T(n)$[35]. The absorption $\alpha$ is given by the ratio of the total oscillator strength and the linewidth of the exciton state. Both scale as $n^{-3}$[10,36] and $\alpha$ remains constant. Therefore, $g_0(n)$ scales as $n^3$ for excitons with a radiatively limited lifetime. In presence of charged impurities, the lifetimes and peak absorption may deviate from the ideal scalings[10,20] and $g_0(n)$ may not show any strong dependence on $n$.

We assume that the reionization of the impurities is mostly driven by the optical field[37,38] and not related to excitons, which yields an intensity-dependent reionization rate:

$$r_{ion} = k_{ion}I,\qquad(5)$$

where $k_{ion}$ is the ionization rate per incoming intensity. Optically driven ionization processes related to impurities are well known[39,40]. Experimentally, we observe that a laser beam does not need to be in

resonance with any exciton state to create an increase of $r_{ion}$, which further supports this assumption. The magnitude of the intensities required shows that the ionization term $k_{ion}$ is significantly smaller than the capture term $\Gamma_{cap}(n)g_0(n)$.

Before analyzing the dynamics in detail, it is worthwhile to check that the model defined above already reproduces the trends observed in experiment. When only one laser beam in resonance with an exciton is present, Eq. (2) yields a steady state value of

$$p_{ion,ss}(n) = \frac{k_{ion}}{k_{ion} + \Gamma_{cap}(n)g_0(n)},\qquad(6)$$

which surprisingly does not depend on laser intensity. When both lasers are active, their capture and ionization rates are added, introducing an intensity dependent competition. In this case, the weak probe laser can typically be neglected and the new steady-state is determined only by the pump laser. This already explains that purification arises only when $n_{pump} > n_{probe}$: Both $\Gamma_{cap}(n)$ and $g_0(n)$ increase with $n$, so $p_{ion,ss}(n_{pump}) < p_{ion,ss}(n_{probe})$ and $\Delta p_{ion} < 0$. The impurities shift towards the neutral state and purification is observed. For $n_{pump} < n_{probe}$, the impurities will instead shift towards a more charged state. This results in enhanced transmission of the probe beam just like blockade does, so these two effects are virtually indistinguishable in the cw measurements reported so far.

So far we have not included any explicit mechanism for the Rydberg exciton-impurity interaction, which determines $\Gamma_{cap}(n)$. Interaction mechanisms on the individual particle level have been studied in detail in atomic physics. Many physical properties of Rydberg states show a characteristic scaling with $n$. This is also true for the interaction cross section in capture events. Each possible interaction mechanism will result in a characteristic $n$-scaling of the interaction cross section[41,42].

Charged impurities are stationary, while the excitons of mass $m_x$ created by the laser move through the crystal with a velocity $v_x$ given by the photon recoil momentum. Their interaction constitutes a capture problem, well known in literature[43]: The classical trajectory of excitons passing by a charged impurity is determined by $v_x$ and the collision parameter $b$, which is the distance between the initial trajectory and a parallel straight line through the center of the impurity, see Fig. 2a. Excitons with small $b$ will be captured, while excitons with large $b$ will be deflected or not affected at all. Upon capture, an exciton might become ionized, neutralizing the impurity while the remaining charge moves to the crystal surface or the exciton might become trapped at the impurity resulting in effective screening[44]. We do not further consider the microscopic product states of the exciton capture process here. For fixed velocity, we can define a collision parameter $b_{max}(n)$ which separates orbits that result in capture and orbits that result in deflection only. Assuming that each capture process neutralizes a charged impurity, we get an interaction cross section $\sigma = \pi b_{max}^2(n)$ that is directly proportional to $\Gamma_{cap}(n)$.

The effective interaction potential consists of two terms. First, the collision has a non-vanishing angular momentum $l = \mu_R v_{coll} b$, where $\mu_R$ is the reduced mass of the exciton and the impurity and $v_{coll}$ is the collision velocity. The angular momentum results in a centrifugal barrier term that reads $V_{cent} = \frac{l^2}{2\mu_R r^2}$, where $r$ is the distance between impurity and exciton. The impurity is much heavier than the exciton and at rest, so $v_{coll} = v_x$ and $\mu_R = m_x = 1.56\,m_0$. The other term corresponds to the second-order Stark shift induced by the electric field $\overrightarrow{E}_{imp}$:

$$\Delta E_{Stark} = -\frac{1}{2}\alpha\,\overrightarrow{E}_{imp}^2,\qquad(7)$$

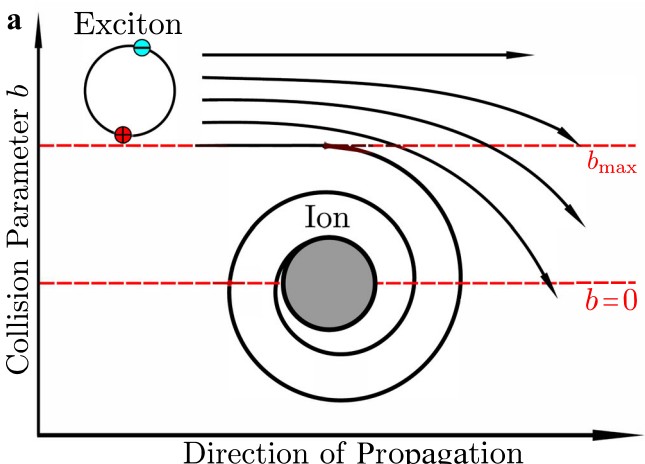

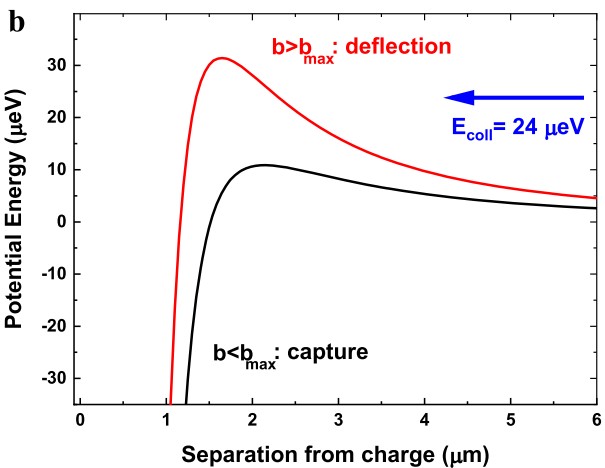

**Fig. 2 | Exciton capture process. a** Schematic depiction of the exciton capture process by a charged impurity. The collision parameter $b$ is given by the distance between the initial trajectory of the exciton and a parallel trajectory passing through the center of the charged impurity. The collision parameter $b_{max}$ divides trajectories that result in capture of the exciton and trajectories that result only in deflection of the exciton. **b** Shape of the total potential composed of the attractive second-order Stark shift and the repulsive centrifugal barrier term for two different collision parameters $b$. The collision energy of $24\,\mu eV$ corresponds to the kinetic energy of the exciton which is determined by the recoil momentum of the photons. An exciton passing the impurity with large $b$ results in a large angular momentum $l$ which in turn increases the centrifugal barrier. The maximum value of the potential is larger than the collision energy, so the exciton becomes deflected. For small $b$, $l$ is lower as well and the centrifugal barrier is reduced significantly. Here, the maximum value of the potential is below the collision energy an the exciton gets captured.

where $\alpha$ denotes the polarizability of the Rydberg exciton. This is the charge-induced dipole interaction which is the strongest interaction between a charge and a neutral composite particle and commonly abbreviated as $\Delta E_{Stark} = -\frac{C_4}{r^4}$ with the long-range interaction coefficient $C_4$ which depends only on $\alpha$ and the dielectric constant of the medium. The sum of both terms yields the effective potential:

$$V_{eff}(r) = \frac{l^2}{2m_x r^2} - \frac{C_4}{r^4}. \tag{8}$$

The shape of this potential is shown in Fig. 2b and has a maximum of $V_{eff}(r_{max}) = \frac{l^4}{16 m_x^2 C_4}$ at the distance $r_{max} = 2\sqrt{\frac{m_x C_4}{l^2}}$[45]. Excitons with collision energies $E_{coll} = \frac{1}{2} m_X v_x^2$ exceeding the effective potential at this point will hit the impurity, while excitons with lower collision energy will be deflected[31]. We then find $l_{max} = 2\sqrt{m_x}(C_4 E_{coll})^{\frac{1}{4}}$ for the largest angular momentum that still passes the centrifugal barrier. The maximum impact parameter resulting in capture is then $b_{max} = \sqrt{2}(\frac{C_4}{E_{coll}})^{\frac{1}{4}}$, which yields the interaction cross section[42]:

$$\sigma = 2\pi \left(\frac{C_4}{E_{coll}}\right)^{\frac{1}{2}}. \tag{9}$$

The interaction coefficient $C_4$ for the induced dipole interaction is mainly determined by the polarizability $\alpha$ of the neutral particle. For Rydberg states, $\alpha$ scales as $\alpha_0 n^7$[13,46,47], where $\alpha_0$ is the ground state polarizability. Therefore, $\sigma$ scales as $n^{3.5}$[43,48,49].

**Data analysis**

As the measurable growth and decay rates are directly proportional to the interaction cross section $\sigma$, we systematically vary the principal quantum numbers of the states we pump and probe in our experiment and identify the interaction in the following. In order to determine the rates, we perform exponential fits both to the onset and decay of purification as shown by the red and green traces in Fig. 1d. We begin with the decay rate $r_2$ of purification as only the probe-induced rates contribute here. We vary the probe beam power, but keep the pump power constant. As an example, a relative transmission series for

$n_{pump} = 16$ and $n_{probe} = 9$ is shown in Fig. 3a. The decay dynamics strongly depend on the probe laser intensity. At probe intensities of $15\,mW/cm^2$ and higher, purification decays within a few microseconds, while it persists up to 70 microseconds at lowest intensities of about $1.2\,mW/cm^2$. At even lower intensities, it can last up to hundreds of microseconds. We repeat this experiment for different $n_{probe}$. In all cases, at low probe intensities, only purification is present and the purification decay rates scale linearly with the probe intensity. An example is shown in Fig. 3b. In the linear regime, each created exciton has the same probability to neutralize a charged impurity by interacting with it during its lifetime. We cut the range of fitted laser powers carefully before any indications for saturation start to occur.

Considering that the capture rates are large compared to the ionization rates, Eq. (4) shows that the slope of the curve we determined in Fig. 3b directly corresponds to $\Gamma_{cap}(n)g_0(n)$. The $n$-scaling of these slopes for different $n_{probe}$ will directly provide us with the required scaling behavior. Figure 3c shows a series of such fits in the linear regime. The data is corrected by a constant offset of $0.025\,\mu s^{-1}$ to highlight the linear power dependence. This offset most likely represents spontaneous changes of the state of the impurities which are not related to the presence of Rydberg excitons. For low $n_{probe}$ below 7, the impurity dynamics are so slow that purification does not fully decay before the arrival of the next pump pulse. Pile-up effects occur, which additionally distort the results. However, the dynamics become drastically faster with increasing $n$ and from $n = 7$ onwards no pile-up occurs. In this range, we can perform a power-law $\propto n^\kappa$ fit to the slopes, as shown in Fig. 3d. We find a power of $\kappa = 6.68 \pm 0.26$, so $\Gamma_{cap}(n)g_0(n)$ scales approximately as $n^{6.5}$.

This scaling behavior contains both the scaling of the Rydberg exciton-impurity interaction $\Gamma_{cap}(n)$ and the scaling of the conversion factor $g_0(n)$ between the optical intensity and the density of excitons. In the purified crystal, the amount of impurities is reduced to a minimum and $g_0(n)$ scales as $n^3$. Thus, the remaining scaling of $n^{3.5}$ is attributed to $\Gamma_{cap}(n)$ and perfectly agrees with the exciton-impurity interaction described above.

We now apply the same analysis to the more complex dynamics at the onset of purification. Here, the capture and purification rates depend on both the pump and probe beams. The rates $r_1$ arising are shown in Fig. 4a for fixed $n_{probe} = 9$ when varying $n_{pump}$ between 10 and 17.

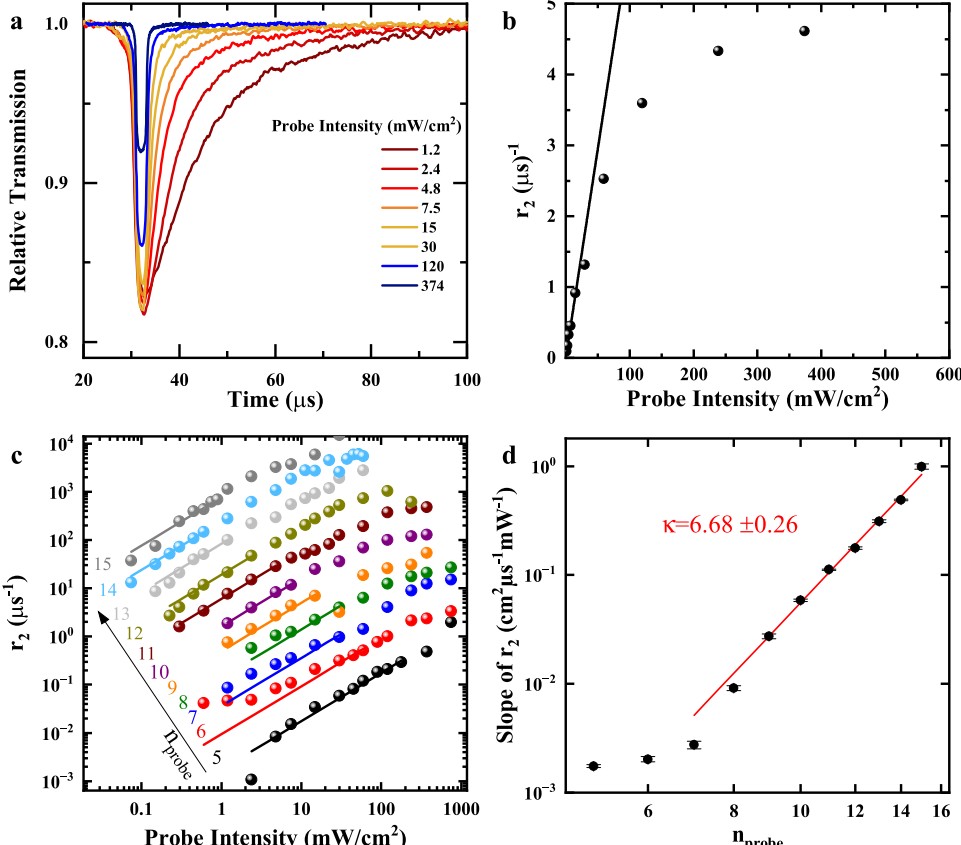

**Fig. 3 | Scaling of decay rate $r_2$. a** Time traces of the relative transmission for $n_{pump} = 16$ and $n_{probe} = 9$. With increasing probe powers the decay of purification becomes significantly faster. **b** Purification decay rate $r_2$ against probe beam intensity for $n_{pump} = 16$ and $n_{probe} = 10$. At low probe intensities, a linear trend arises which saturates at higher intensities. The solid line denotes a linear fit to the low-intensity region. **c** Probe-power series of $r_2$ for pumping at $n_{pump} = 16$ and varying $n_{probe}$ from 5 to 15. The pump intensity is adjusted such that the highest possible purification is reached for each $n_{probe}$. Solid lines are linear fits to the linear regions. The data points and lines are offset vertically and plotted on a double-logarithmic scale for clarity. A small offset of $r_2 \approx 0.025\,\mu s^{-1}$ is subtracted from all curves. **d** Power law fit to the slopes of the purification rate $r_2$. The error bars are obtained by varying the range of fitted data points in panel (**c**) by ±1. For $n_{probe} > 6$, the scaling is close to $n^{6.5}$.

Initially, the results seem qualitatively comparable to the case of $r_2$. The rates show a linear increase with pump intensity for low enough intensities. When extrapolating the slopes to vanishing pump intensity, a finite rate of approximately $1.05\,\mu s^{-1}$ remains, which corresponds to the contribution of the probe beam intensity to the capture and ionization processes. The data shown in panel a is corrected by this offset to visualize the linear trend. We determine the slopes of the rates and again perform a power law fit. The results are shown in Fig. 4b. Surprisingly, the results do not show the scaling observed for $r_2$. The data scales as $n^{3.96\pm0.26}$ from $n = 10$ to 17, see gray dashed line. For $n_{pump}$ above 12 the slopes of $r_1$ are close to a scaling of $n^{3.5}$, which is shown by the red line. For lower $n_{pump}$ the scaling is significantly stronger, but we do not have enough data points to determine whether the scaling can be described by a single power law or whether it flattens continuously.

While this result may at first glance suggest different mechanisms at the onset and decay of purification, we propose the following explanation within our model: Experimentally, we determine $\Gamma_{cap}(n)g_0(n)$, the joint scaling of the impurity-exciton interaction and the conversion factor between the pump intensity and the exciton density created. The scaling for $n_{pump}$ above 12 is very close to the $n^{3.5}$-scaling of the charge-induced dipole interaction. The conversion factor $g_0(n)$, on the other hand, crucially depends on the exciton lifetime. In addition to the standard decay channels via phonon interactions and radiative decay, purification itself can affect it, which in turn depends on the impurity states. When the pump beam arrives, the number of impurities in the charged state is

so large that for any Rydberg exciton is more likely to decay by capture at an impurity than by recombination. Therefore, $g_0(n)$ does not show any scaling with the principal quantum number and the observed scaling of the slope of $r_1$ is given solely by $\Gamma_{cap}(n)$. We estimate the $n$-dependence of $g_0(n)$ in presence of charged impurities from the transmission spectrum shown in Fig. 1a, where no pump laser is present and the amount of impurities is high. The inset in Fig. 4b shows $g_0$ normalized to $g_0(10)$. Indeed, $g_0(n)$ varies only slightly within the range from $n = 10$ to 17 and does not contribute to the $n$-scaling of the rate $r_1$. Thus, the observed scaling is in agreement with the $n^{3.5}$-scaling expected for the bare charge-induced dipole interaction, in particular for $n > 12$. This strongly supports the identification of the long-ranged charge induced-dipole interaction as the main mechanism of the interaction between Rydberg excitons and charged impurities. Only at low $n$ below 12, the natural lifetime of the Rydberg excitons again becomes so short that excitons may decay without encountering an impurity. This is reflected by the steeper slope in this region.

We thus anticipate two general scenarios, where the mean time it takes for an exciton to encounter a charged impurity is either shorter or longer than its natural lifetime. At the end of the pump pulse, the number of charged impurities takes its minimal value. Here, it is likely that Rydberg excitons decay without encountering an impurity, so the $n^3$-scaling of $g_0(n)$ again enters the scaling of the slope of $r_2$. The observed differences in the scaling of the slopes of $r_1$ and $r_2$ therefore directly represent the influence of purification.

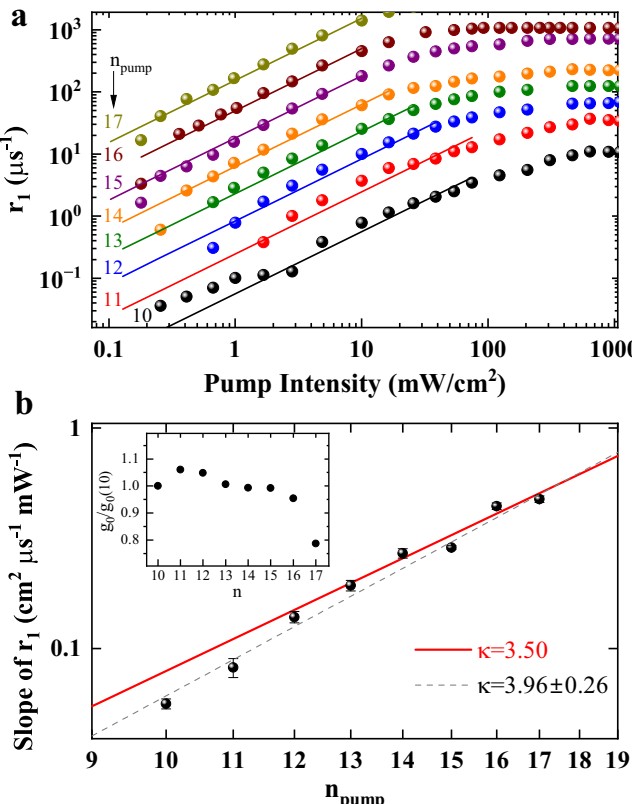

**Fig. 4 | Scaling of growth rate $r_1$. a** Pump-power series of $r_1$ in time-resolved relative transmission in a pulsed pump-probe experiment for probing at $n_{probe} = 9$ and varying $n_{pump}$ from 10 to 17. The probe intensity amounts to 7.5 mW cm$^{-2}$. Solid lines are linear fits to the linear regions. The data points and the lines are offset vertically and plotted on a double-logarithmic scale for clarity. All curves are corrected by a constant offset of 1.05 $\mu$s$^{-1}$. **b** The red lines shows a scaling with $n^{3.5}$ according to the model. For large $n$ above 12, the data fits to this prediction well. For lower $n$ the scaling becomes steeper. A fit to the whole range of principal quantum numbers yields $n^{3.96\pm0.26}$. The error bars are obtained by varying the range of fitted data points in panel a by $\pm 1$. The inset shows $g_0$ estimated from the linear absorption spectrum. The variation of $g_0$ is below 10% between $n = 10$ and 16.

## Optimal excitation energy

We have already demonstrated that Rydberg excitons are a good tool to neutralize impurities. However, there are only few material systems that open up the possibility to selectively excite individual Rydberg exciton states. This raises the question whether our findings are relevant for other material systems as well. To this end, we perform another cw pump-probe experiment. The probe beam is in resonance with $n_{probe} = 12$ and we scan the pump beam across the entire Rydberg exciton spectrum up to beyond the band gap. While scanning the pump laser energy, the probe laser absorption coefficient $\alpha_{12}$ becomes enhanced or reduced compared to its unperturbed value $\alpha_{12,0} \approx 48.5$ mm$^{-1}$. Panel b in Fig. 5 shows the changes in the probe absorption coefficient when scanning the pump laser, while panel a shows the linear absorption spectrum from Fig. 1a, so we can directly compare all relevant influences on the probe beam absorption.

At pump laser energies below $n_{pump} = 12$, the presence of the pump laser reduces the probe laser absorption coefficient. We note that this is also true when the pump laser is not in resonance with any exciton state. This is a consequence of indirect absorption into 1S states via the continuous phonon background. Indeed, the probe absorption directly follows the shape of the phonon background as shown in Fig. 5c. In a cw experiment, 1S excitons may accumulate over time and decay via an Auger-process into a hot electron-hole plasma.

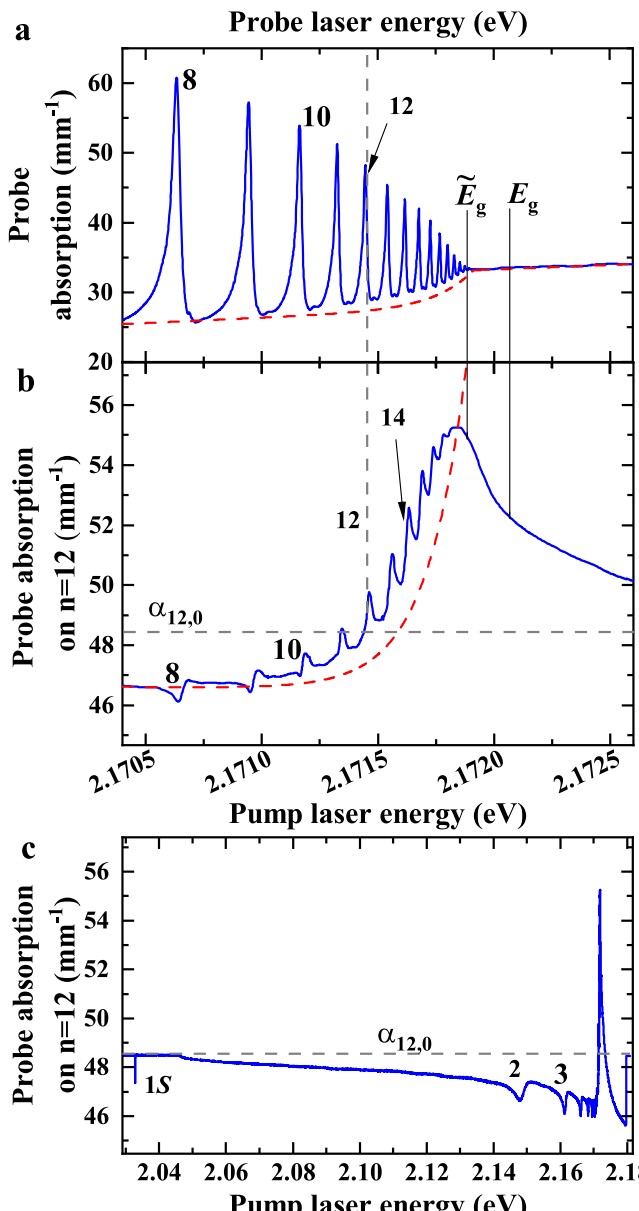

**Fig. 5 | Spectral dependence of the purification effect. a** The linear absorption spectrum from Fig. 1a in the energy range from $n = 8$ to the band gap, shown as a reference. The red dashed line indicates the background that is dominated by an exponential tail below the apparent band gap $\tilde{E}_g$. **b** Probe laser absorption coefficient as a function of pump laser energy. The probe laser energy is fixed to the peak of the $n_{probe} = 12$ exciton resonance, while the pump laser energy is scanned. The pump laser enhances the absorption of $n_{probe} = 12$ if its energy is set within a spectral range around the apparent band gap $\tilde{E}_g$. The grey dashed vertical line marks the resonance energy of $n = 12$, the grey dashed horizontal line marks the absorption coefficient of $n = 12$ without the presence of a pump laser $\alpha_{12,0}$. Panel (**c**) shows an extended energy range of panel (**b**).

The plasma, in turn, can directly cause the reduced probe absorption as it leads to a broadening of absorption lines[30]. However, with increasing pump laser energy the probe absorption coefficient $\alpha_{12}$ increases again starting from approximately $n_{pump} = 11$ onwards (panel b). It grows continuously and eventually overcomes $\alpha_{12,0}$ when $n_{pump}$ crosses $n_{probe}$. This trend continues up to a pump laser energy of $E_{max} = 2.17187$ eV, where we observe the largest enhancement of the probe laser absorption coefficient. This energy corresponds to the apparent band gap $\tilde{E}_g$, which is the energy of the highest Rydberg

states that can still exist and above which the continuum of free electrons and holes begins. At pump energies above $\tilde{E}_g$, $\alpha$ decreases again. Here, the above-band-gap excitation initially creates a low-energy electron-hole plasma, which may also cause purification directly or indirectly via relaxation into high-$n$ Rydberg states. At the same time, these free charge carriers will also create additional stray fields, which result in the observed reduction of purification.

In the spectral range directly below the band gap, the probe spectrum shows an exponentially increasing background absorption (panel b). This continuous increase can be described by an Urbach-like exponential tail $\sim \exp((E - \tilde{E}_g)/E_U)$ with $E_U \approx 170\ \mu eV$, which is exactly the same spectral width required to describe the exponential tail below the band gap in the linear absorption spectrum (panel a), as indicated by the dashed line. This exponential tail lies on top of the phonon background and it contains some discrete Rydberg exciton states. However, the high-$n$ Rydberg states are highly sensitive to tiny local fields and perturbations and therefore may show significant energy shifts comparable to the energy spacing between consecutive states. In this case, they form an inhomogeneously broadened peak close to the apparent band gap $\tilde{E}_g$. Interestingly, it is this region where purification is most efficient, although it is not possible to identify individual Rydberg states anymore. We conclude that purification does not require excitation of a certain well-defined Rydberg exciton state, but a mixture of high-$n$ states is fully sufficient. This means that in some technologically relevant semiconductor materials it may be possible to excite such mixtures of highly excited Rydberg states by spectrally narrow pumping directly below the band gap and to achieve purification, even though these materials do not show resolvable individual Rydberg exciton states.

## Discussion

In conclusion, we demonstrated that the interaction between Rydberg excitons and charged impurities is long-ranged and governed by a $C_4$ charge-induced dipole interaction. We have shown that we can significantly reduce the number of charged impurities and their detrimental effects by injecting Rydberg excitons into the crystal. We observe an absorption enhancement of up to 25% and find that purification may persist at least up to hundreds of microseconds. Blockade instead is found to act on time scales much faster than 100 ns. Our model directly explains the scaling laws of the observed purification rates. It further implies that the purification level is given by the $n$ of the predominant Rydberg exciton and therefore explains why purification is observed only when $n_{pump} > n_{probe}$. The observed enhancement of absorption is only the additional effect of the pump beam which adds to the purification already caused by the probe beam. Therefore, our findings also explain the surprisingly low impurity density of less than $0.01\ \mu m^{-3}$ found in earlier reports[20,27,30]: As these experiments employed continuous cw probe beams, the impurity density observed was actually the density of charged impurities in an already partially purified crystal. An interesting question concerns the highest resolvable principal quantum number: Without additional illumination, the density of charged impurities defines the highest observable principal quantum number $n_{max}$[20]. In the presented experiments, additional laser illumination reduces the number of charged impurities through purification, yet an increase of $n_{max}$ is not observed. We explicitly discuss this effect in Ref. [50]: Possibly, excess charges created during the purification process act as a low-density plasma, cause a reduction of absorption on high-$n$ states and, thus, counteract an increase of $n_{max}$. In other words, purification may effectively serve to turn static defect charges into mobile charges, thus making the crystal more homogeneous but not completely charge-free.

Although a thin plasma could also be expected to cause purification, we observe such an effect only for pump laser energies slightly above the band gap. Our hypothesis is that at higher pump laser energies, the plasma kinetic energies and associated temperatures are too high for effective purification to occur. A detailed investigation of the complex microscopic interaction between a plasma, Rydberg excitons and impurities constitutes an intriguing outlook for future research. Conductivity measurements might shed light on the microscopic process of the impurity neutralization. Any residual charges created by the neutralization process should then become detectable. Our results represent an important step towards controlling detrimental charge noise, will elevate the role of Cu$_2$O as a platform for quantum technologies[19,51–57] and may open up the possibility to perform ultracold chemistry-like analogy experiments[58] in semiconductors.

## Methods
### Experimental setup
We study the time-resolved relative transmission of a natural crystal with a thickness of $d = 30\ \mu m$ that is not oriented along a high symmetry axis at 1.38 K. We apply an electro-optical modulator to create pump beams with a duration of $\tau_{pulse} = 2\ \mu s$ out of the cw emission of a tuneable dye laser. The repetition rates are $f = 5\ kHz$ for probe power series and 30 kHz for pump power series. In case of positive purification, Fig. 1e, we used $f = 15\ kHz$. The rise and fall times of the modulator are 15 ns each. The cw probe beam is provided by another tunable dye laser and its energy is fixed to a specific exciton resonance. We detect the probe beam transmission with a streak camera. To capture the purification dynamics, the time windows are set to 10 μs or 100 μs. μ The time resolution amounts to about 100 ns in the first case and to 1 μs in the latter case. We measure the average power $P_{avg}$ in front of the cryostat with an accuracy of $0.01\ \mu W$. Losses by reflection of 4% at three cryostat windows (six surfaces) and 25% at the first sample surface reduce the incoming power by a factor of ~0.59. We calculate the peak intensity per pulse by $I_{peak} = 0.59 P_{avg}/(f \tau_{pulse} \pi r_{beam}^2)$. $r_{beam}$ is the laser beam radius at the sample position. The intensity can be determined with an accuracy of about 0.25 mW/cm$^2$ for pump powers and 0.1 mW/cm$^2$ for probe powers. The pump and probe beam diameter amount to 220 μm and 100 μm FWHM, respectively. Both beams show a linewidth of approximately 1 neV and can be tuned into resonance with any Rydberg exciton state. Pump and probe beam polarizations are orthogonal and the probe beam transmission is polarization-filtered to reject pump light scattered into the probe beam path. Additionally, the probe beam is filtered further using a pinhole. The probe beam power amounts to 1 μW, which converts to an intensity of 7.5 mW/cm$^2$, unless noted otherwise. In the cw experiments (Fig. 5), both lasers are unmodulated. The probe laser energy is set into resonance with the $n = 12$ Rydberg exciton while the pump laser energy is scanned continuously. We detect the transmitted light $I$ of the probe laser as a function of pump laser energy and calculate the absorption coefficient $\alpha = -\frac{1}{d}\ln(I/I_0)$. We obtain the bare probe laser absorption of the $n = 12$ resonance of 48.5 mm$^{-1}$ by blocking the pump laser and a comparison with a linear absorption spectrum, e.g. Fig 1a. The probe laser absorption coefficient with respect to the pump laser's energy directly reflects the change of the peak absorption at $n = 12$ induced by the pump laser. Values smaller (larger) than that mean a decrease (increase) of absorption.

### Data fitting
To determine the interaction cross-section of the purification process we use a linear regression as depicted in Fig. 3c and Fig. 4a erring towards lower intensities to avoid the possible inclusion of Rydberg blockade related effects. This method is probed for robustness by varying the fit range within ± 1 data points, which defines the error bars in Figs. 3d and 4b.

**a**

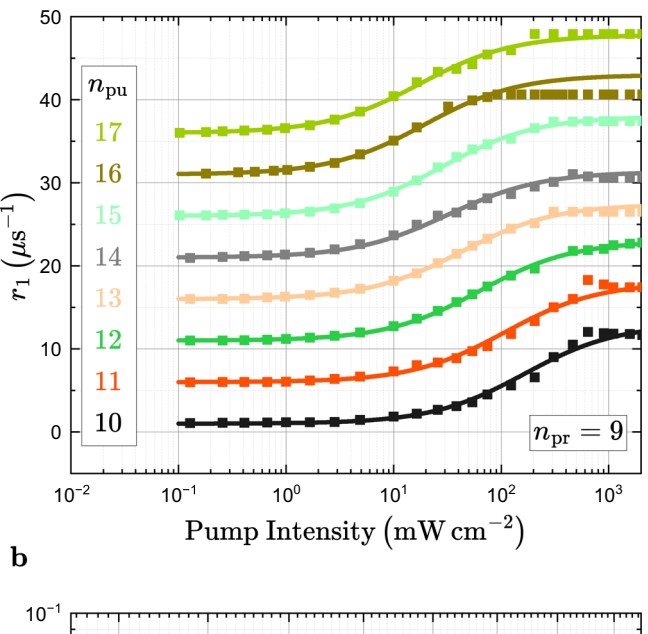

**b**

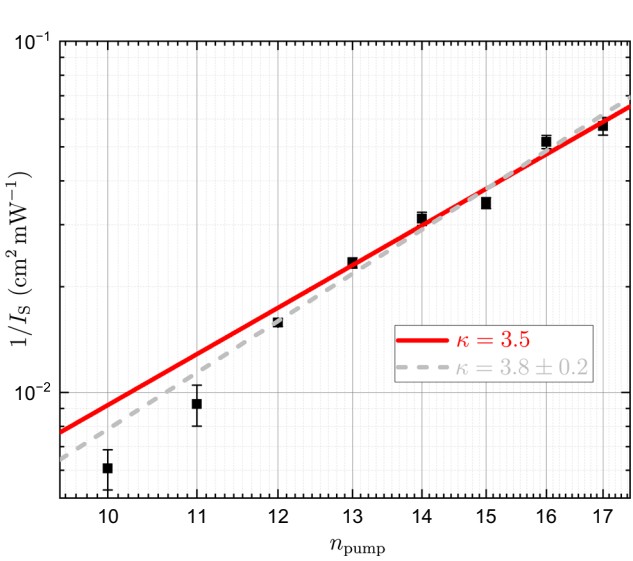

**Fig. 6 | Alternative fit routine for $r_1$. a** Alternative fit routine using a sigmoid curve over the complete measured intensity range. This approach includes all phenomena, purification as well as Rydberg blockade. **b** Alternative slope values with corresponding numerical errors from the fits in panel (**a**). The results remain well within the theoretical expectation with slightly steeper progression towards lower $n_{pump}$.

**a**

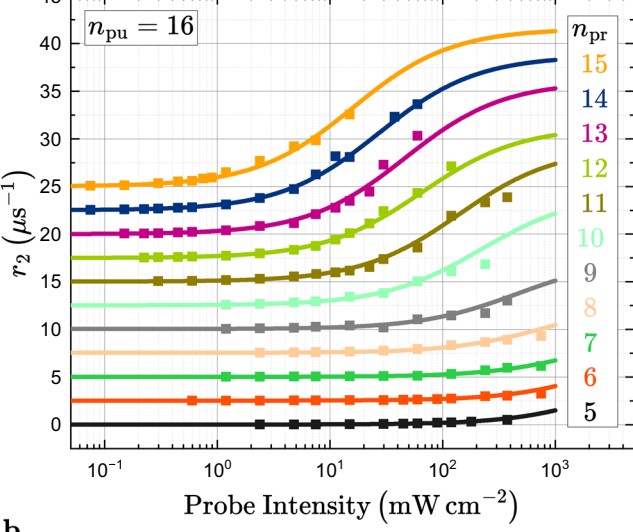

**b**

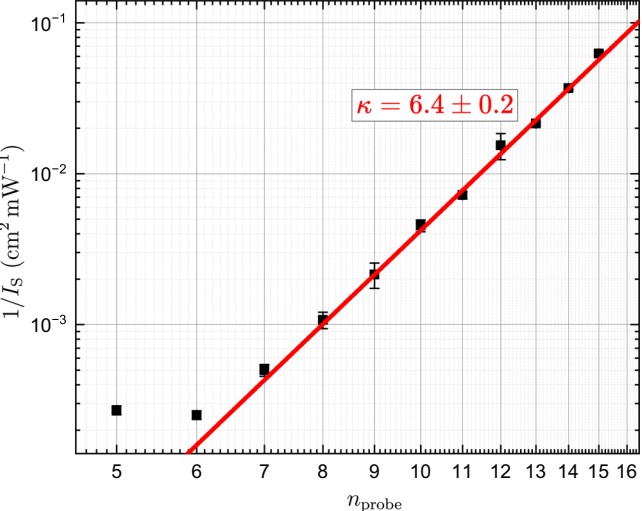

**Fig. 7 | Alternative fit routine for $r_2$. a** Alternative fit routine using a sigmoid curve over the complete measured intensity range. Due to the fact that only the decay of the purification process is considered it is safe to assume that no or very little Rydberg blockade is included over the complete intensity range. The changes in the rates, however, are very small towards lower $n_{probe}$, which results in a large uncertainty for $r_A$. **b** Alternative slope values with corresponding numerical errors from the fits in panel (**a**). There results are again very well within the expected range and also in agreement with the linear regression method.

An alternative approach has also been considered, where a nonlinear fit function of the form:

$$r(I) = r_0 + r_A \frac{\frac{I}{I_S}}{1 + \frac{I}{I_S}}, \quad (10)$$

is used.

This function forms a sigmoid curve with the lower limit $r_0$ and upper limit $r_A$, where the former depends on the CW probe-laser and the latter generally reflects the time resolution of the setup. The results are depicted in Figs 6 and 7 and do not differ from the linear regression method used in the main text. However, this method is inconsistent, as we discuss in the following. For the pump intensity series (Fig. 6a) the complete sigmoid curve is visible in the data, making it a perfect match for the function. This means, however, that both phenoma are

included in the fit, i.e. purification and Rydberg blockade, which is not the case for the linear regression method. The probe intensity series (Fig. 7a) only includes the decay of purification over the complete measurement range and is therefore less likely to include Rydberg effects. In this case the change of the rates, that occurs within the range of measurements, are too small to describe the full sigmoid curve. This renders the determination of the value for $r_A$ difficult and its value, in turn, affects the remaining parameters.

The most important fit parameter is $I_S$, which is akin to the reverse of the slope value of the linear regression method, although it is less intuitive to seek the proportionality to the interaction cross-section in this manner. The values obtained with the sigmoid curve are depicted in Figs. 6b and 7b. It is noteworthy that, although the interpretation for Fig. 6a is ambiguous due to the inclusion of the Rydberg blockade, the

same results as those obtained by a linear regression are extracted from the values for $I_S$. The results for the other data set are also within theoretical considerations and in agreement with the linear regression method.

In summary, both methods yield comparable results. The linear regression methods is more consistent as long as the linear regime is considered and data already affected by either time resolution or Rydberg blockade is excluded from the fit.

## Data availability
The datasets generated during and/or analysed during the current study are available from the corresponding author on request.

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

## Acknowledgements

We gratefully acknowledge support of this work by the Deutsche Forschungsgemeinschaft through grant number 504522424 and 316133134 (M.B., B.P., M.H., S.S., J.H. and M.A.). V.W. acknowledges support by the NSF through a grant for the Institute for Theoretical Atomic, Molecular, and Optical Physics at Harvard University and the Smithsonian Astrophysical Observatory. We kindly thank Heinrich Stolz for helpful scientific discussions and for coining the term purification.

## Author contributions

M.A. and J.H. designed the experiment. M.B., B.P., M.H., S.S. and J.H. performed the experiments. M.A. and V.W. developed the theory. M.B. and J.H. analyzed the data. All authors wrote and commented on the manuscript.

## Funding

## Competing interests

The authors declare no competing interests.
