## [Peer Review File · Nature Communications]

REVIEWER COMMENTS

Reviewer #1 (Remarks to the Author):

The authors describe in their manuscript a pump-probe experiment with which they show how to saturate charged impurities in semiconductor materials supporting excitonic Rydberg states. Their argument relies on the observation that, by preceding a probe laser to excite a particular Rydberg exciton by a pump laser pulse generating a different Rydberg excitation, the absorption of the probe laser increases. This is explained by a mechanism that saturates the charged impurities by breaking up the Rydberg excitons from the first (pump) laser. The theoretical model, simple as it might seem, does explain the observed scaling laws very well. The simplicity of the model in fact advantageous as it gives a clear meaning without too restrictive assumptions.

Despite the clarity of the explanations, I do have a few questions regarding some of them.

From line 189 onwards, the focus is on larger pump than probe excitons. The reason for that becomes clear only much later. It would be helpful to the reader if either a preliminary explanation or a pointer towards the paragraph after line 275 would be included.

In line 206, the density of impurities is estimated to be below $1(\text{micron})^3$ which is based on the assumption that the Debye model is valid for this excitonic system. In their Ref.[46], this assumption is being questioned. How does the density estimate change if a correlated plasma model is being used?

In the paragraph starting at line 212, the authors consider the mechanism of an exciton breaking up, thereby neutralizing an impurity, with the remaining charge moving to the sample surface. Should these surface not be detectable? And if not, why not?

I also spotted a few linguistic errors that need correcting:

line 177: 'sophisticated' -> 'characteristic'?

lines 89, 102, 191: 'principle quantum number' -> 'principal quantum number'

line 361: 'hundredth' -> 'hundreds'?

In conclusion, my view is that the manuscript provides novel insights into the behaviour of highly excited (Rydberg) excitons in cuprous oxide and paves the way to coherent manipulation of Rydberg excitons. I can therefore recommend publication in Nature Communications once the authors have answered my questions.

Reviewer #3 (Remarks to the Author):

This paper presents a detailed study of the interaction of Rydberg excitons with charged defects in Cu₂O. The results are impressive, the authors show that the presence of a low power pump beam resonant with a highly excited Rydberg states can lead to substantial (around 25%) improvements in the visibility of high-n Rydberg states. They term this effect “purification” and the proposed interpretation is that Rydberg excitons are neutralising charged impurities in the system. A thorough study of the n dependence of the effect is given, with the most effective energy for purification begin observed just above the region where Rydberg states are no longer observed.

The authors provide a theoretical model based on a classical picture of an exciton being trapped at charged impurities and derive predicted scaling laws form this model. The scaling laws of such a process are derived and are compared to the experimentally measured values. There is reasonable agreement between the two, though we have a few concerns with the comparison which we detail in the comments below.

Overall, the paper is of high-quality and is clearly written. The effect studied is of interest to those studying Rydberg excitons and to the wider semiconductor field, as it probes the interaction of excitons and charged impurities. Additionally, this paper may be of interest to the atomic physics community, where the interactions between Rydberg states and point defects have been extensively studied. We recommend the paper for publication as long as the following comments are addressed:

Major points

Figures/results

Figure 1(b) is the figure where purification first appears, but the figure and its description on line 111 are unclear. On the y axis, differential transmission is presumably $(I_{\text{on}} - I_{\text{off}})/I_0$ where I_{on} and I_{off} are the transmitted intensities with and without the pump beam respectively, not I/I_0 as indicated (I/I_0 can never be negative). Concerning the x-axis, it is not clear which photon energy is fixed and which is scanned. We assume that the pump energy is fixed on resonance with $n=16$, and that the probe is scanned? In this case the axis should be labelled as probe photon energy (or perhaps more usefully as detuning). The text should be modified to make it clearer how one goes from 1(a) to 1(b).

A major issue with this and many other figures is that the colour scheme is ambiguous – similar colours are used for widely different intensities. A “monotonic” colour scale (eg varying from grey through yellow orange to red) would help a lot, as would showing fewer lines. In the legend, are three significant figures justified for the intensity? What is the uncertainty in the intensity?

Figure 3(c) and 4(a) It is not clear from the text and these figures what condition was used to determine the extent of the “linear” region, or how robust the extracted quantities in Figures 3(d) and 4(b) are to this choice. For example, for the $n_{\text{probe}}=11$ data in 3(c), it looks like a straight line with a much lower gradient could be used to fit the data up to probe intensities of around 50 mW/cm^2 , and for $n=8-10$ a linear fit would appear to be a good model across the whole range. Using a logarithmic intensity scale may make the behaviour in the low-intensity region clearer. As the gradients from these fits give the power law scaling in 3(d) and 4(b) it is crucial that it is clear how the data was fitted, including how the errorbars in 4(b) depend on the range of data included.

An improved analysis would be to fit the data with a phenomenological saturation model (e.g. $r_2 = (I/I_S)/(1+I/I_S)$), and then to extract the initial gradient from the fit. This would have the advantage of using all the available data for each curve in a way that does not require introducing a cut-off. Since strong conclusions are drawn from the value of $\kappa = 6.5$ fitted to figure 3(d), addressing these issues is crucial.

Figure 4(b) The blue line appears to be fitted to just three points, and therefore the uncertainty on the gradient is not statistically meaningful as there are not enough degrees of freedom. This power law should not be quoted unless more data can be fitted. It would also be interesting to discuss how much the gradient of the red curve (and its uncertainty) vary as more points are added to the fit. Is it unreasonable to fit all of the points in figure 4(b) with a single power law? How would this look on the plot?

A better approach here might be to use the theory to make a prediction and constrain the power law to $\kappa=3.5$. This would enable the reader to judge the degree of agreement in a way that is independent of the number of points included in the fit.

Text

Free carriers and plasmas. Throughout the text effects related to free carriers and electron-hole plasmas are discussed, but it is hard to get an overall sense of their impact and the relation to the other effects discussed here (blockade and purification). This aspect of the article needs to be made significantly clearer, perhaps by explicit consideration of free carriers/plasma formation as a 3rd mechanism alongside blockade and purification from the start? Specific issues are

- Line 27: It is argued that free carriers cannot saturate the defects due to spatially inhomogeneous Stark shifts. If the free carriers are created throughout the laser spot, why can they not achieve the same result as excitons? Alternatively does purification process not suffer from the same spatial inhomogeneity? (see for example the above-gap purification in Fig. 5)

- Line 470 Discussion of figure 5(b): Here there is a discussion about how the absorption is reduced when the pump laser is below $n=12$ but is not on resonance with an exciton state. This is attributed to indirect absorption resulting in a plasma. However, earlier in the article the reduction in absorption is attributed to solely to Rydberg blockade. The authors need to clarify the relation between these two effects and justify or modify the description in terms of blockade used earlier.

- Line 491: The potential for electron-hole plasma to screen the charged defects is only superficially discussed. A further discussion for this should be added, including potential reasons for why the plasma does not cause a similar purification effect.

Scaling of widths with n . At multiple points in the paper (e.g. line 257) it is stated that the purification means that the exciton widths now follow their expected n^{-3} scaling laws, which is important to get the good agreement between the predicted and measured scaling laws. However, the scaling of the widths with and without purification is not shown. These should be added and discussed.

Line 283 Discussion of capture model: The capture model for Rydberg states at charged impurities is based on well-known atomic physics models. This link to previous work is mentioned in the text but should be made clearer, for example, reference [43] should appear nearer where the model is introduced. Additionally, as this model is not unique to this work, we are not sure figure 2 is necessary.

Line 380 Uncertainty and significant figures for scaling laws. Quoted values should have uncertainties; these are sometimes present on the figures but should appear in the text too.

Line 435: "This unambiguously identifies..." We disagree with this. We think the observed scaling laws are compatible with this process, but we do not think the attribution can be described as unambiguous. Additionally, adding more of the lower n points to the fit in figure 4(b) may change the scaling law (see discussion above).

A discussion about the limits to the Rydberg series should be added. Does the purification increase how many states can be resolved? If not, why not?

Minor points

- Line 89 "Principal" not "principle" on lines: 89, 102, 191, 249 check throughout
- Line 11 repetition on line 11 and 12, second sentence of the paragraph is very similar to the first
- Line 61, "studied" not "at study here"
- Line 95 repetition in paragraph beginning at line 95, discussion on highest n is repeated from earlier
- Line 120 is it not the state width that matters rather than the laser linewidth?
- Line 142 say how narrow the lasers are
- Line 143 "different" not "difference"
- Line 167 the stated value is an intensity not a power (text refers to power)
- Line 179 The units for the quoted intensity have a typo.
- Line 180 could be clearer when discussing the interplay between blockade and purification, more references to the curve on figure 1(d) would help.
- Line 249 I_n is introduced near line 249 but isn't used later. In equation 4 it is referred to as I.
- "a" and "\alpha" are both used for absorption coefficient, "a" is used on lines 462, 468, 489.
- Figure 5(b) Scale on 5(b) is unreadable as the inset obscures one of the axis labels; y scale and label should be added to the inset.
- Line 568 the stated value is a power not an intensity.

Reviewer #4 (Remarks to the Author):

Review NCOMMS-23-19988

In this article, the authors experimentally and theoretically explore a mechanism to “purify” the spectra of Rydberg excitons in Cu₂O by neutralizing charged crystal impurities. This neutralization comes from the capture of mobile charges (presumably provided by the breakup of excitons) by fixed impurities. The purification being in competition with Rydberg blockade in CW experiments, the authors use a time-resolved pump-probe approach to separate the two effects. The purification alone leads to a 25% increase in the height of the excitonic absorption peaks. While interesting, substantial and useful, this result doesn't look like a game-changer for the field - unless it can be shown to also significantly increase the maximum principal quantum number achievable in a given sample, which doesn't seem to be the case.

The experiment is well thought and very well executed. The theory is interesting and explains nicely most of the results, although I wonder if the exciton-only interpretation is enough. The paper itself is clear and well written. I only have some questions and a few minor comments. Therefore, I would recommend publication in Nature Communication after the authors address the following questions/remarks.

Major remarks:

1. The big question is: does this method improve the highest achievable Rydberg state in a given sample, n_{max} ? After all, even if the pump has to be at a higher energy than the probe, the largest purification happens at the gap energy, which is significantly above n_{max} for most samples (typically by more than the energy separation between n_{max} and $n_{max} + 1$, or even higher). Therefore, it could possibly increase n_{max} for the probe. So, can it?
 - a. If yes, this would largely change the impact this paper can have as it would address a major issue, namely the supply of (effective) high-quality crystals. Therefore, the authors should test and/or mention this. But, as this is an obvious question that the authors completely ignore, I guess this is not the case?
 - b. If not, then why? From the current literature, one would expect that what limits n_{max} is precisely the stray E-fields suppressed here.
 - c. In any case, this question should be explicitly addressed in the paper.

2. It would be useful to show somewhere the best enhancement this method can provide to a typical Rydberg series spectrum, by adding a figure comparing probe spectra with and without the pump placed at the optimal energy (at the effective gap) and at the optimal power. This “best probe” case could for example be added to fig.1a.
3. I wonder what kind of charged defects are neutralized here. Are these Cu and/or O vacancies? Other types of impurities? Are they deep defects (largely below-gap energy) or shallow defect (energy just below the gap)?
4. Actually, neutralizing shallow defects with below-gap laser excitation is a well-known way to avoid stray fields in semiconductor systems, notably quantum dots, and the fact that the best purification is not found on a clearly defined Rydberg state but at the effective gap is strongly reminiscent of shallow defects neutralization. Isn't such shallow defect addressing also visible here? While I don't doubt the exciton description is correct, it may not be the whole picture...
5. Why are the authors only considering immobile defects?
6. Why are the authors assuming that only excitons are captured by charged impurities and not some low-energy free charges? As the authors note at the end of the paper, at all pump energies there is some (non-negligible) phonon-assisted creation of 1S excitons that create some e-h plasma via Auger effect. Such charges should also be captured and neutralize charged defects, likely before they have a detrimental effect on Rydberg excitons via screening. It seems possible this could have an important impact, maybe even participate in the difference between the r_1 and r_2 scaling? Actually, Fig.5b (which I find really interesting) clearly shows that having the pump resonant with a Rydberg state has a limited effect: the phonon background has a dominant effect on the magnitude of the purification, especially at higher energies, providing efficient purification even when not addressing any exciton. This seems important and I wonder if this could be quantitatively estimated and/or added to the theory?
7. On a related note, the purification seems to continue quite far into the gap in in figure 5b: the $n = 12$ absorption is increased up to 2176 meV, 4-5 meV above the gap. This is very large by Rydberg excitons standards. There, the explanation in term of the pump creating an ill-defined mixture of high Rydberg excitons does not hold. A low-energy plasma does, reinforcing the previous point. If

only excitons participated, I would expect a much sharper drop into the gap. Could the authors please comment?

8. On yet another related note, in the fig. 5b inset we see that there is some positive purification from the background (negative purification from excitons) for the low pump energies. More precisely, the phonon background seems to purify ($a > a_0$) for pump energies approximately lower than $n=4$. How does that fit into the model / the authors' explanation?
9. Following up on the previous point, what happens at lower energies still? If the low energy purification is indeed entirely due to the phonon background inducing a plasma via $1S+Auger$, it should brutally stop below 2.047 eV. It would be interesting to show and discuss such data.

Minor remarks:

- a. Typo on page 2, line 143. "difference" should be "different"
- b. Typo on page 2, line 179. mW/cm^{-2} should be mW/cm^2 or $mW.cm^{-2}$
- c. Typo on page 2, line 191. "principle" should be "principal"
- d. Typo on page 5, line 301. "effected" should be "affected"
- e. I guess from context that " $r_{1(2)}$ " is $1/\tau_{1(2)}$, but if I am not mistaken this is written nowhere. It would be nice for the reader to either use $1/\tau_{1(2)}$ or define $r_{1(2)}$ explicitly somewhere.
- f. I find the inset of fig.5b really interesting and I would make it bigger, possibly into a fig.5c.

Reviewer #1 (Remarks to the Author):

The authors describe in their manuscript a pump-probe experiment with which they show how to saturate charged impurities in semiconductor materials supporting excitonic Rydberg states. Their argument relies on the observation that, by preceding a probe laser to excite a particular Rydberg exciton by a pump laser pulse generating a different Rydberg excitation, the absorption of the probe laser increases. This is explained by a mechanism that saturates the charged impurities by breaking up the Rydberg excitons from the first (pump) laser. The theoretical model, simple as it might seem, does explain the observed scaling laws very well. The simplicity of the model in fact advantageous as it gives a clear meaning without too restrictive assumptions.

Despite the clarity of the explanations, I do have a few questions regarding some of them.

From line 189 onwards, the focus is on larger pump than probe excitons. The reason for that becomes clear only much later. It would be helpful to the reader if either a preliminary explanation or a pointer towards the paragraph after line 275 would be included.

We added a pointer to the clarifying discussion in the next Section in line 214.

In line 206, the density of impurities is estimated to be below $1(\text{micron})^3$ which is based on the assumption that the Debye model is valid for this excitonic system. In their Ref. [46], this assumption is being questioned. How does the density estimate change if a correlated plasma model is being used?

We thank the Referee for this question and want to clarify the way impurity densities are estimated in the different models.

Within the Debye model, Ref. [27] (now), the impurity density was estimated to be on the order of $0.01 \text{ 1}/\mu\text{m}^3$ as a result from the non-zero bandgap shift at zero pump power. There, no explicit model for the impurities was considered.

In Ref [30] (former Ref. [46]), it was found that the plasma has a dominant contribution to the change in absorption spectra at densities of $0.01 \text{ 1}/\mu\text{m}^3$ and above. At pump powers leading to lower plasma densities, the impurities have a major contribution. Therefore, the density of impurities could be estimated to be even below $0.01 \text{ 1}/\mu\text{m}^3$.

The best estimate for an impurity density, however, can be found in Ref [20], where the impact of the presence of charged impurities on the absorption spectra is explicitly discussed. Here, an even lower value of $0.001 \text{ 1}/\mu\text{m}^3$ is found.

We decided to cite that paper in line 236 instead of Ref [27].

Our assumption being below a density of $1/\mu\text{m}^3$ holds in all three cases.

We also changed the sentence in line 566 correspondingly.

In the paragraph starting at line 212, the authors consider the mechanism of an exciton breaking up, thereby neutralizing an impurity, with the remaining charge moving to the sample surface. Should these surface [charges] not be detectable? And if not, why not?

The Referee is fully correct. Residual charges that move to the surface should in principle be detectable. One possibility is to contact the crystal surface directly via electrical wires. Unfortunately, any direct contact to the sample surface leads to strain among the sample which is likely to disturb the system and hinder the high-n exciton creation. Assuming a density of $0.001 \text{ 1}/\mu\text{m}^3$ we expect about 2000-3000 impurities within the illuminated volume. Any measurable current will be on the order of Femtoampere which renders its detection highly nontrivial. So far such a measurement was not done but will be considered in future experiments. We added a comment in the discussion (line 590). Another way to detect single charges at the surface is to measure the current at the tip of an atomic force microscope (AFM). Unfortunately, such a device is not available in the institute at the moment.

I also spotted a few linguistic errors that need correcting:

line 177: 'sophisticated' -> 'characteristic'?

lines 89, 102, 191: 'principle quantum number' -> 'principal quantum number'

line 361: 'hundredth' -> 'hundreds'?

Thank you for spotting these typos. We corrected all of them.

In conclusion, my view is that the manuscript provides novel insights into the behaviour of highly excited (Rydberg) excitons in cuprous oxide and paves the way to coherent manipulation of Rydberg excitons. I can therefore recommend publication in Nature Communications once the authors have answered my questions.

Reviewer #2

Reponse to Question 1 and 2:

We thank the Referee for the important remarks. They address an important point, which is the change of n_{max} . In this context we performed the following measurements so far:

To study the change in the absorption spectrum when purification is most efficient, we measured several CW-absorption spectra with a pump laser energy set to the energy where the increase of absorption was largest. This energy was found by a pump laser scan similar to the one shown in Fig. 5b in the manuscript. In that particular measurement, this energy was found to match with the energy of the highest Rydberg states, i. e., an energy slightly below the apparent band gap \tilde{E}_g . This is shown in Figure 1 in this response letter. So far, we did not set the pump laser to an even higher energy. However, already at this pump laser energy no increase of absorption can be observed for the highest states with $n = 20, 21$, see lower panel of Fig. 1.

We also tested a sample of lower quality with $n_{max} = 14$. This is shown in Figure 2 below. In this sample, we find purification to be most efficient at an energy higher than n_{max} . This energy is indicated by the red arrow in the inset of Figure 2. However, even here, purification only affects the exciton lines that are visible without pump laser, but does not increase n_{max} .

Figure 1

Figure 2

A thorough analysis of CW-absorption spectra including the data shown in Figs.1 and 2 is under preparation as a follow-up publication (cited now as Ref. [50]). Therefore, we did not plan to show the data in the present manuscript.

As the Referee states, from the current literature [20 now] a suppression of electric stray fields should increase the number of visible states. In the following we discuss possible reasons why this is not observed.

Although the presence of charged impurities has a large effect on the Rydberg excitons, the model described in Ref. [20] does not reproduce all features observed in the experimental spectra. We assume that the presence of charged impurities is not the only detrimental effect and additional mechanisms have to be considered, in particular when a pump laser is added to the system. In case of purification, the residual charges may, for example, not move to the surface and form a low-density plasma instead. These mobile charges may neutralize impurities, but will also dynamically screen the system and counteract the purification effect. The latter effect will reduce the absorption of the highest Rydberg states instead. If a low-density plasma is formed during the neutralization process or if the residual charges move to the crystal surface will be considered in future conductivity experiments, as discussed in the response to Referee 1.

We added a discussion about the change of n_{max} and a reference to our follow-up manuscript in the conclusions, from line 569 onwards.

3)

I wonder what kind of charged defects are neutralized here. Are these Cu and/or O vacancies? Other types of impurities? Are they deep defects (largely below-gap energy) or shallow defect (energy just below the gap)?

We thank the referee for this question and try to make this point clearer.

In Cu_2O there are Cu and O vacancies. The Cu vacancies are not charged and most prominent in artificial crystals. In natural crystals, oxygen vacancies are dominant. Oxygen vacancies are found in a singly (O^+) or doubly charged state (O^{++}), see for example Refs. [18] and [19]. With binding energies of about 500 meV and more they can be considered as deep defects. Indications of metallic impurities can be found in photoluminescence measurements as well around 200 meV below the band gap [44], but their charge state is not well defined. To the best of our knowledge, clear indications for shallow donor states were not found so far.

We added this information in line 39 ff.

4)

Actually, neutralizing shallow defects with below-gap laser excitation is a well-known way to avoid stray fields in semiconductor systems, notably quantum dots, and the fact that the best purification is not found on a clearly defined Rydberg state but at the effective gap is strongly reminiscent of shallow defects neutralization. Isn't such shallow defect addressing also visible here? While I don't doubt the exciton description is correct, it may not be the whole picture... In the present manuscript we focus on the exciton-impurity interaction. If there are shallow impurities present, we may neutralize them by below-gap excitation. Taking this effect into

account is an interesting task for future investigations.

However, illuminating the sample with infrared light does only cause small changes in the Rydberg exciton spectra which we attribute to photoionization of impurities. If shallow impurities with small binding energies were the dominant species in the material, we would expect larger changes in the spectra.

We further refer to the answer on question 6.

5)

Why are the authors only considering immobile defects?

Beside static defects, also mobile vacancies that hop between lattice sites can be present in Cu_2O . Reports on both copper and oxygen vacancies can be found in literature. Copper vacancies are found to diffuse faster than oxygen vacancies, while only the latter are typically charged. To the best of our knowledge the diffusion lengths are negligible on the timescales of the pump pulse lengths, which renders the problem quasi static. Diffusion constants reported in literature at about 400K are on the order of $10^{13} \mu\text{m}^2/\mu\text{s}$, see Ref. [1] of this response, At lower temperatures they might be even smaller.

6)

Why are the authors assuming that only excitons are captured by charged impurities and not some low-energy free charges? As the authors note at the end of the paper, at all pump energies there is some (non-negligible) phonon-assisted creation of 1S excitons that create some e-h plasma via Auger effect. Such charges should also be captured and neutralize charged defects, likely before they have a detrimental effect on Rydberg excitons via screening. It seems possible this could have an important impact, maybe even participate in the difference between the r_1 and r_2 scaling?

The Referee points on an interesting question, whether free charges neutralize charged defects as well. We do not exclude an impurity neutralization by free charges. The effect is likely since the Coulomb attraction between free charges and impurities is strong.

However, an e-h plasma created by Auger effect of indirectly excited 1S excitons is hot and the carriers created via Auger effect are of high kinetic energy which reduces the capture probability. Moreover, the presence of free charges also leads to detrimental effects like dynamical screening of the excitonic system and a band gap renormalization which counteracts purification and renders its observation in the absorption spectrum cumbersome.

A low energy plasma does indeed purify, which can be observed for pump energies slightly above the band gap in Fig. 5b. We noted that in lines 514ff.

However, free electrons and holes are detrimental charges, too, and it seems reasonable to assume that no purification can be observed by creating a spatially homogeneous e-h plasma unless the probability for each charge to encounter an impurity is higher than 50%. For the low defect densities present in our samples, this is likely never the case. However, the situation may be different in very bad samples with a huge number of detrimental charged impurities.

Coming back to question 4 raised by the Referee, selective excitation of free carriers solely in the immediate spatial vicinity of charged impurities indeed could be advantageous. This can in principle be achieved by pumping very slightly below the band gap. The electric field created by the charged impurity may then result in a Franz-Keldysh-like reduction of the band gap close to the impurity, which renders absorption into free carriers possible. At the same time, the absorption spectrum of high- n excitons will change due to both the reduction in the band gap and the presence of the electric field, which renders the exciton spectrum less well defined. Indeed, such effects may contribute to what constitutes purification around the band gap region. However, this will be a non-trivial mixture of excitons, some not that well defined shifted states that constitute the Urbach tail and possibly low-energy unbound, but Coulomb correlated electrons and holes, which will also undergo relaxation to other states. Unraveling this complicated mixture will be a highly interesting topic for future studies, but it is unfortunately out of the scope of the present manuscript.

Actually, Fig.5b (which I find really interesting) clearly shows that having the pump resonant with a Rydberg state has a limited effect: the phonon background has a dominant effect on the magnitude of the purification, especially at higher energies, providing efficient purification even when not addressing any exciton. This seems important and I wonder if this could be quantitatively estimated and/or added to the theory?

The Referee states that the phonon background has a dominant effect on the magnitude of purification. We would like to emphasize that we divide the background into phonon-assisted absorption into $1S$ excitons and an Urbach-like absorption tail close to the band gap. Only the latter causes purification. The tail is caused by overlapping high- n Rydberg exciton lines that are broadened and mixed by the electric field of impurities. Here, a mixture of excitons with high principal quantum numbers n but also angular momentum quantum numbers is created. Therefore, we always address excitons by pumping into this part of the background. However, the density and actual composition of exciton states excited by pumping this background becomes complex and hard to control. An inclusion into the model is beyond the scope of the present manuscript, but a challenging task for future investigations. Therefore, we focus on the simpler case in which we resonantly excite well defined exciton states. We tried to emphasize the different physical origins of phonon background and Urbach tail more in line 528.

7)

On a related note, the purification seems to continue quite far into the gap in in figure 5b: the $n = 12$ absorption is increased up to 2176 meV, 4-5 meV above the gap. This is very large by Rydberg excitons standards. There, the explanation in term of the pump creating an ill-defined mixture of high Rydberg excitons does not hold. A low-energy plasma does, reinforcing the previous point. If only excitons participated, I would expect a much sharper drop into the gap. Could the authors please comment?

As we state in the response to question 6, a low-energy plasma that is excited by excitation directly above the bandgap can purify impurities as well. With increasing energy above the gap the kinetic energy increases which reduces the capture probability of the free carriers and the purification amplitude with increasing energy.

Moreover, one could think about a formation of Rydberg excitons directly from the low-energy plasma which is more likely at low energies.

8)

On yet another related note, in the fig. 5b inset we see that there is some positive purification from the background (negative purification from excitons) for the low pump energies. More precisely, the phonon background seems to purify ($a > a_0$) for pump energies approximately lower than $n=4$. How does that fit into the model / the authors' explanation?

We think our figure was misleading and added a horizontal line indicating the value of α_0 .

Moreover, the former inset became a separate panel c now. The data was remeasured to extend the energy scale to lower energies. A comparison with the horizontal line shows that there is no purification caused by absorption into the phonon background at energies around $n=4$.

9)

Following up on the previous point, what happens at lower energies still? If the low energy purification is indeed entirely due to the phonon background inducing a plasma via $1S+Auger$, it should brutally stop below 2.047 eV. It would be interesting to show and discuss such data.

We think the question arises from the assumption that absorption into the phonon background causes purification and conclude that Figure 5b was misleading so far. We restate that we optimized the figure by adding a reference line and absorption into the phonon background does not cause purification.

Nevertheless, an excitation at photon energies below the $1S$ exciton or in the infrared range was tested and shows small purification-like changes in the absorption spectrum with magnitudes which are much smaller than the changes observed when pumping at excitons. We conclude that the change in probe laser absorption at low laser powers becomes negligible for pump energies below the energy of 2.047 eV, i.e. where the phonon background starts.

To clarify this point, we remeasured the data in Fig5 b and c and extended the energy range to values below the $1S$ state. Indeed, the effect caused by the phonon background – which leads to a reduction of absorption – stops around 2.047 eV.

Minor remarks

We corrected all minor remarks and typos. The conversion between rates and time is stated in lines 251 ff.

We changed the inset of figure 5 to a separate panel c.

Reviewer #3 (Remarks to the Author):

This paper presents a detailed study of the interaction of Rydberg excitons with charged defects in Cu₂O. The results are impressive, the authors show that the presence of a low power pump beam resonant with a highly excited Rydberg states can lead to substantial (around 25%) improvements in the visibility of high-n Rydberg states. They term this effect “purification” and the proposed interpretation is that Rydberg excitons are neutralising charged impurities in the system. A thorough study of the n dependence of the effect is given, with the most effective energy for purification begin observed just above the region where Rydberg states are no longer observed.

The authors provide a theoretical model based on a classical picture of an exciton being trapped at charged impurities and derive predicted scaling laws form this model. The scaling laws of such a process are derived and are compared to the experimentally measured values. There is reasonable agreement between the two, though we have a few concerns with the comparison which we detail in the comments below.

Overall, the paper is of high-quality and is clearly written. The effect studied is of interest to those studying Rydberg excitons and to the wider semiconductor field, as it probes the interaction of excitons and charged impurities. Additionally, this paper may be of interest to the atomic physics community, where the interactions between Rydberg states and point defects have been extensively studied. We recommend the paper for publication as long as the following comments are addressed:

Major points

Figures/results

Figure 1(b) is the figure where purification first appears, but the figure and its description on line 111 are unclear. On the y axis, differential transmission is presumably $(I_{on} - I_{off})/I_0$ where I_{on} and I_{off} are the transmitted intensities with and without the pump beam respectively, not I/I_0 as indicated (I/I_0 can never be negative. Concerning the x-axis, it is not clear which photon energy is fixed and which is scanned. We assume that the pump energy is fixed on resonance with $n=16$, and that the probe is scanned? In this case the axis should be labelled as probe photon energy (or perhaps more usefully as detuning). The text should be modified to make it clearer how one goes from 1(a) to 1(b).

The Referees are fully correct. We corrected the x-axis title to “probe photon energy (eV)”. The y-axis is corrected to differential transmission is $\Delta I = I_{on} - I_{off}$, i.e. the difference between the transmitted intensities with and without pump.

A major issue with this and many other figures is that the colour scheme is ambiguous – similar colours are used for widely different intensities. A “monotonic” colour scale (eg varying from grey through yellow orange to red) would help a lot, as would showing fewer lines. In the legend, are three significant figures justified for the intensity? What is the uncertainty in the intensity?

We thank the Referees for their suggestion. We reduced the number of visible lines and changed the color scheme to a monotonic one in Figure 1 and 3 ranging from red to blue.

The values given in the legend stem from a conversion of applied pump powers to intensities, which leads to somewhat odd numbers. We discuss the conversion between pump power and intensity in the methods section. We measure the power with an accuracy of 10 nW, which translates to about 0.25 mW/cm^2 for the pump power. We rounded the given intensities accordingly.

Figure 3(c) and 4(a) It is not clear from the text and these figures what condition was used to determine the extent of the "linear" region, or how robust the extracted quantities in Figures 3(d) and 4(b) are to this choice. For example, for the $n_{\text{probe}}=11$ data in 3(c), it looks like a straight line with a much lower gradient could be used to fit the data up to probe intensities of around 50 mW/cm^2 , and for $n=8-10$ a linear fit would appear to be a good model across the whole range. Using a logarithmic intensity scale may make the behaviour in the low-intensity region clearer. As the gradients from these fits give the power law scaling in 3(d) and 4(b) it is crucial that it is clear how the data was fitted, including how the errorbars in 4(b) depend on the range of data included. We thank the Referees for this comment and admit that the representation of data was disadvantageous. We chose a logarithmic intensity scale in Figures 3(b) and (c) and 4(b) now to highlight the behavior at low pump powers. Moreover, the full dots represent the data that was taken into account for fitting and open dots indicate unused data.

We checked the robustness of the fit by varying the data range by +/- one data point and calculated the error bars from the deviation of the resulting slopes. We added a comment in the figure captions and a description in the methods section. The error bars have been updated in panel 3(d) and 4(b).

The fitting ranges were cut at low powers before any saturation effects occur that could lower the slopes. This is now mentioned in the text from line 388 on.

An improved analysis would be to fit the data with a phenomenological saturation model (e.g. $r^2 = r(I/I_S)/(1+I/I_S)$), and then to extract the initial gradient from the fit. This would have the advantage of using all the available data for each curve in a way that does not require introducing a cut-off. Since strong conclusions are drawn from the value of $\kappa = 6.5$ fitted to figure 3(d), addressing these issues is crucial.

We thank the Referees for the suggested alternative fit function, which fits the data visually very well over the complete measurement range of Fig. 4a), but the complete range represents very different processes depending on the intensity. All phenomenological regions are included in the fit, dominant purifying, intermediate phase and dominant Rydberg blockade. Thus it is (very) questionable to include higher intensity data points in the fits for Fig. 4a). On the other hand, Fig. 3c) does not suffer from this limitation, since the time scale of the decay of the purification is measured. However, in this case the fit parameter r , which is the upper saturation value can only be guessed from the data's trend.

Nevertheless, something akin to the slope values in Fig. 3d) and 4b) can be obtained from the inverse of the fit parameter I_s . It is less intuitive than the slope value for analysis, which can be expected to contain a value proportional to the interaction cross section. The results of this analysis are still in agreement with our original analysis underlying the robustness of the data. This alternative fit function is discussed in the methods section now.

Figure 4(b) The blue line appears to be fitted to just three points, and therefore the uncertainty on the gradient is not statistically meaningful as there are not enough degrees of freedom. This power law should not be quoted unless more data can be fitted. It would also be interesting to discuss how much the gradient of the red curve (and its uncertainty) vary as more points are added to the fit. Is it unreasonable to fit all of the points in figure 4(b) with a single power law? How would this look on the plot?

Purification occurs only when the pump beam excites a state with higher principal quantum

number than the probe beam. In order to test many different pump n we probed at a low principal quantum number, i.e. $n=9$. Probing at even lower n was avoided since the change in α becomes negligible small. Consequently, there is no data for lower pump n than $n=10$ ($9+1$). As the Referee states, the scaling of $n^{3.4}$ depends on the range of principal quantum numbers included in the fit. To test that, we also fitted all points in 4(b). This is shown by the green line in Figure 3 in this file below.

Figure 3

We obtain an overall exponent of 3.9, which lies in between the values of 3.4 and 5.4, but beyond of the errorbars that were given so far. Therefore, we decided to change the plot as described in the following.

A better approach here might be to use the theory to make a prediction and constrain the power law to $\kappa=3.5$. This would enable the reader to judge the degree of agreement in a way that is independent of the number of points included in the fit.

We thank the Referees for this suggestion. In order to uncouple the obtained scaling law from the number of points included in the fit, we removed the blue line in the main text and show the scaling with $n^{3.5}$ predicted by the model as a solid red line. The data points at low n deviate from this curve, which we interpret as a steeper scaling in the text (lines 426 ff).

Moreover, we show the fit to all datapoints by a dashed grey line.

Text

Free carriers and plasmas. Throughout the text effects related to free carriers and electron-hole plasmas are discussed, but it is hard to get an overall sense of their impact and the relation to the other effects discussed here (blockade and purification). This aspect of the article needs to be made significantly clearer, perhaps by explicit consideration of free carriers/plasma formation as a 3rd mechanism alongside blockade and purification from the start?

We thank the Referees for this remark and tried to introduce the plasma models used so far in literature and discuss the current understanding of the plasma in an own paragraph in the

introduction from line 65 onwards. The effect of a plasma on the interaction between excitons and impurities is not well understood so far and its investigation beyond the present manuscript. We also comment on the impact of a plasma now in the discussion (line 583 ff).

Specific issues are

- Line 27: It is argued that free carriers cannot saturate the defects due to spatially inhomogeneous Stark shifts. If the free carriers are created throughout the laser spot, why can they not achieve the same result as excitons? Alternatively does purification process not suffer from the same spatial inhomogeneity? (see for example the above-gap purification in Fig. 5)

We thank the Referees for this remark. We would like to point out that we do not intend to say that free carriers cannot saturate defects. They certainly can. We intended to point out that for zero-dimensional systems, such as quantum dots, only the electric field at the very position of the QD matters. Here, one wants to avoid telegraph noise and therefore wants to have a charge configuration that is stable in time. However, the most stable charge configuration is not necessarily a neutral one. Our intention was to point out that stable configurations carrying a net charge will still result in spatially inhomogeneous Stark shifts, which renders them problematic for non-zero-dimensional systems.

Changing a neutral impurity charge configuration to a charged one during an encounter with an exciton will be a neutral-neutral interaction with an activation energy corresponding to the exciton binding energy, which at least renders such an interaction less likely compared to an encounter between the neutral impurity configuration and a free charge.

However, if a neutral defect state is the most stable one, indeed free carriers should be highly efficient in neutralizing them.

- Line 470 Discussion of figure 5(b): Here there is a discussion about how the absorption is reduced when the pump laser is below $n=12$ but is not on resonance with an exciton state. This is attributed to indirect absorption resulting in a plasma. However, earlier in the article the reduction in absorption is attributed to solely to Rydberg blockade. The authors need to clarify the relation between these two effects and justify or modify the description in terms of blockade used earlier. We thank the Referees for this remark and extended the discussion about possible interaction mechanisms in the introduction by the plasma blockade in line 65 onwards.

- Line 491: The potential for electron-hole plasma to screen the charged defects is only superficially discussed. A further discussion for this should be added, including potential reasons for why the plasma does not cause a similar purification effect.

We fully agree that the discussion of the impact of an electron-hole plasma has to be extended. We decided to add a paragraph in the discussion from line 583 on.

For more details we would like to refer to our response to question 6 of Referee 2.

Scaling of widths with n . At multiple points in the paper (e.g. line 257) it is stated that the purification means that the exciton widths now follow their expected n^{-3} scaling laws, which is important to get the good agreement between the predicted and measured scaling laws. However, the scaling of the widths with and without purification is not shown. These should be added and discussed.

We thank the Referee for this important remark.

In former line 257 (now line 277) the word linewidth was written. Here, we meant the homogeneous linewidth which corresponds to the radiative lifetime, which determines the conversion factor g and the exciton density.

Unfortunately, the linewidths in CW spectra can be inhomogeneously broadened and do not give access to the radiative lifetimes. We changed the wording from "linewidth" to "lifetime" to avoid any confusion. We thank the Referee for making this point clear

In case of purification the number of impurities is reduced to a minimum such that it is unlikely for an exciton to encounter an impurity and its lifetime is radiatively limited rather than limited by a collision with an impurity. The scaling of the radiative lifetime $\propto n^3$ therefore enters the rate r_2 again.

We fully agree, that it is an interesting task to study the change of linewidths in CW spectra. We plan a follow-up publication where we thoroughly analyse the CW response to purification (Ref. [50]). This is also mentioned in our response to questions 1 and 2 of Referee 2. For completeness, we show the results here in Figure 4 below for both linewidth (left) and oscillator strengths (right).

Figure 4

We find that the linewidths indeed decrease and scale closer to the ideal scaling. However, at high n we still observe some amount of inhomogeneous broadening in CW spectra.

We plan to employ a 2D spectrometer in future investigations to access the homogeneous linewidth directly.

Line 283 Discussion of capture model: The capture model for Rydberg states at charged impurities is based on well-known atomic physics models. This link to previous work is mentioned in the text but should be made clearer, for example, reference [43] should appear nearer where the model is introduced. Additionally, as this model is not unique to this work, we are not sure figure 2 is necessary.

The Referees are fully correct. Reference [43] (updated) is cited now in line 315, where the capture model is introduced.

We would like to keep Figure 2 as we think that this kind of capture model is not well known in the semiconductor community.

Line 380 Uncertainty and significant figures for scaling laws. Quoted values should have uncertainties; these are sometimes present on the figures but should appear in the text too.

We thank the Referees for this remark. We added error bars in the text.

Line 435: "This unambiguously identifies..." We disagree with this. We think the observed scaling laws are compatible with this process, but we do not think the attribution can be described as unambiguous. Additionally, adding more of the lower n points to the fit in figure 4(b) may change the scaling law (see discussion above).

We replaced the word "unambiguously" by "supports" and attribute this statement specifically to the high-n range.

A discussion about the limits to the Rydberg series should be added. Does the purification increase how many states can be resolved? If not, why not?

We thank the Referees for this important question. We studied the effect of purification on cw spectra as well. Interestingly, an increase of the maximum principal quantum number is not observed, neither in samples of high quality nor of lower quality. We kindly refer to our answer to question 1 from Referee 2, where we show this in Figures 1 and 2 of this response letter. We believe that any additional charges introduced into the system by pump laser cause a bleaching of highest exciton states which counteracts on purification. This is discussed in the discussion section of the manuscript from line 569 onwards.

Minor points

We thank the Referees for stating the following minor points from which we corrected all.

- Line 89 "Principal" not "principle" on lines: 89, 102, 191, 249 check throughout. Corrected.
- Line 11 repetition on line 11 and 12, second sentence of the paragraph is very similar to the first. Removed sentence one of the introduction.
- Line 61, "studied" not "at study here". Corrected.

- Line 95 repetition in paragraph beginning at line 95, discussion on highest n is repeated from earlier We removed the repetitive statement about the highest principal quantum number being \$n=30\$.

- Line 120 is it not the state width that matters rather than the laser linewidth?

We replaced "laser linewidth" by "state linewidth"

- Line 142 say how narrow the lasers are.

We now mention the laser linewidth of 1 neV in that paragraph. It is also given in the methods section.

- Line 143 "different" not "difference" Corrected.

- Line 167 the stated value is an intensity not a power (text refers to power). Text refers to intensity now.

- Line 179 The units for the quoted intensity have a typo. Corrected.

- Line 180 could be clearer when discussing the interplay between blockade and purification, more references to the curve on figure 1(d) would help.

We thank the Referee for this remark. We tried to make the passage in lines 194 ff. (former line 180) clearer and tried to refer to the different curves given in panel (d) more often.

- Line 249 I_n is introduced near line 249 but isn't used later. In equation 4 it is referred to as I .

We refer to intensity as \$I\$ throughout the text now.

- "a" and " α " are both used for absorption coefficient, "a" is used on lines 462, 468, 489.

We apologize and refer to the absorption coefficient as \$\alpha\$ now.

- Figure 5(b) Scale on 5(b) is unreadable as the inset obscures one of the axis labels; y scale and label should be added to the inset.

The inset became a separate panel (c) now.

- Line 568 the stated value is a power not an intensity.

The power is converted to an intensity.

Reviewer #4 (Remarks to the Author):

We kindly thank the referee for co-reviewing. We are highly aware that taking part in peer-review is a time-consuming and sometimes unrewarding process. Rigorous peer review nevertheless is a cornerstone for ensuring the high standards in science. Therefore, we are especially grateful to

each early career researcher who takes the responsibility to maintain the standards of the field and participates in peer review.

[1] A. Mittiga, F. Biccari and C. Malerba, Intrinsic defects and metastability effects in Cu_2O , Thin Solid Films **517** (2009) 2469–2472

REVIEWERS' COMMENTS

Reviewer #1 (Remarks to the Author):

The authors have answered my questions and amended their manuscript my satisfaction. I regards this study as an important milestone in our understanding of highly excited Rydberg excitons, and I am happy to see the it published in Nature Communications.

Reviewer #2 (Remarks to the Author):

In this revised version of the manuscript and the rebuttal, the authors satisfactorily answered all of my questions. As far as I can see, that authors also did a good job at addressing the other referees' comments and questions.

I am now happy with the current version of the manuscript and therefore recommend publication.

Reviewer #3 (Remarks to the Author):

The authors' response and the revised manuscript show that the authors have comprehensively responded to all the points raised in our original report.

We are therefore happy foe the article to proceed to publication.

However we make two suggestions to cosmetically improve figures 3(a) and 4(a)

1. The authors should consider a logarithmic y axis in addition to the x-axis, to emphasise the slope in the low-intensity region and to make it appear linear

2. The solid lines are fits to the low intensity region only. For clarity, these lines should not be plotted beyond the linear region where they are valid. This would draw the reader's eye to the region of agreement, rather than the high-intensity region where they disagree.

Reviewer #4 (Remarks to the Author):

Point-by-Point response to the reviewers' comments

Reviewer #3 (Remarks to the Author):

However we make two suggestions to cosmetically improve figures 3(a) and 4(a)

1. The authors should consider a logarithmic y axis in addition to the x-axis, to emphasise the slope in the low-intensity region and to make it appear linear

We thank the Referees for this useful suggestion. We assume that the Referees mean 3(c) instead of 3"(a)", since it shows similar content as 4(a). Therefore, we present the data shown in figures 3(c) and 4(a) now in a double-logarithmic way, as the referees suggested.

2. The solid lines are fits to the low intensity region only. For clarity, these lines should not be plotted beyond the linear region where they are valid. This would draw the reader's eye to the region of agreement, rather than the high-intensity region where they disagree.

Also, the linear fits are shown in the relevant data range only. Minor changes in the text were necessary to describe these changes. All changes are highlighted in an additional pdf named "TRPurifying_ReSub_JH_red.pdf"